# Integration of questionnaire-based risk factors improves polygenic risk scores for human coronary heart disease and type 2 diabetes

Max Tamlander [1], Nina Mars[1], Matti Pirinen [1,2,3], FinnGen*, Elisabeth Widén[1] & Samuli Ripatti [1,2,4✉]

Large-scale biobank initiatives and commercial repositories store genomic data collected from millions of individuals, and tools to leverage the rapidly growing pool of health and genomic data in disease prevention are needed. Here, we describe the derivation and validation of genomics-enhanced risk tools for two common cardiometabolic diseases, coronary heart disease and type 2 diabetes. Data used for our analyses include the FinnGen study (N = 309,154) and the UK Biobank project (N = 343,672). The risk tools integrate contemporary genome-wide polygenic risk scores with simple questionnaire-based risk factors, including demographic, lifestyle, medication, and comorbidity data, enabling risk calculation across resources where genome data is available. Compared to routinely used clinical risk scores for coronary heart disease and type 2 diabetes prevention, the risk tools show at least equivalent risk discrimination, improved risk reclassification (overall net reclassification improvements ranging from 3.7 [95% CI 2.8–4.6] up to 6.2 [4.6–7.8]), and capacity to be improved even further with standard lipid and blood pressure measurements. Without the need for blood tests or evaluation by a health professional, the risk tools provide a powerful yet simple method for preliminary cardiometabolic risk assessment for individuals with genome data available.

[1] Institute for Molecular Medicine Finland, FIMM, HiLIFE, University of Helsinki, Helsinki, Finland. [2] Clinicum, Department of Public Health, University of Helsinki, Helsinki, Finland. [3] Department of Mathematics and Statistics, University of Helsinki, Helsinki, Finland. [4] Broad Institute of MIT and Harvard, Cambridge, MA, USA. *A list of authors and their affiliations appears at the end of the paper. ✉email: samuli.ripatti@helsinki.fi

Approaches to utilize genome data in healthcare have been increasingly investigated during recent years, resulting in development of genetic prediction algorithms with a major focus on the prevention of common complex diseases such as coronary heart disease (CHD), type 2 diabetes (T2D), and breast and prostate cancer[1–5]. Several genetic testing platforms now allow calculation of polygenic risk scores (PRS)[2], which combine the effects of numerous genetic markers across the genome to a single measure of genetic risk[6]. PRSs are able to stratify people effectively to distinct trajectories of future disease risk[4,5], but their utility is limited when used in the absence of other known risk factors.

Clinical practice guidelines for the prevention of CHD and T2D advocate risk calculators to estimate the 10-year risk of disease, to identify and target individuals at high risk[7–10]. The risk is calculated with clinical risk factors such as sex, age, smoking, and blood lipid levels. In case of high total risk, individuals are offered preventative medication and encouraged to adhere to a healthy lifestyle. Current clinical risk calculators are, however, failing to identify up to 40% high-risk individuals[7,11], and as prevention has broader benefits than treating diseases, improving identification of at-risk individuals is an important public health challenge.

Although CHD and T2D are known to be highly heritable[12,13], current risk prediction tools do not utilize directly measured genetic information. CHD and T2D develop predominantly as a combination of unfavorable lifestyle and hereditary factors[14,15], with effects accumulating over the course of life. Identifying high-risk individuals using both genetic and non-genetic risk factors at an early stage for targeted preventative efforts could therefore have substantial benefits over the current prevention strategies for CHD and T2D. As a growing number of individuals have genome data available, one potential approach for improving risk estimation is to utilize PRSs together with simple online questionnaires to preselect people from the population for further comprehensive clinical risk evaluation. Here, we (1) show that PRSs combined with simple and easily surveyable risk factors, including demographic and lifestyle factors, and comorbidities, provide a viable tool to identify high-risk individuals in CHD and T2D, (2) show that Genomics-enhanced RIsk Tools (GRIT) combining genome-wide risk and these simple risk factors for CHD and T2D (GRIT-CHD and GRIT-T2D) have at least comparable performance to risk scores advocated by clinical guidelines, and (3) show that adding standard lipid and blood pressure measurements to our GRIT scores leads to notable performance improvements over current clinical risk scores for CHD and T2D. We derive our risk tools in the Finnish biobank study, FinnGen, and externally validate them in an independent cohort, the UK Biobank.

## Results

**Study characteristics and GRIT derivation.** First, we built genome-wide PRSs for CHD and T2D by obtaining weights from the largest genome-wide association studies (GWASs) on European-ancestry individuals that do not overlap with UK Biobank[12,13]. We tested the association of these PRSs in FinnGen ($N = 309,154$ Finnish individuals) with 33,628 cases of CHD and 44,266 cases of T2D and in the UK Biobank ($N = 343,672$) with 18,698 cases of CHD and 24,192 cases of T2D. We observed improved performance over previously published scores[4] (Supplementary Table 1), and therefore chose our $PRS_{CHD}$ and $PRS_{T2D}$ built with PRS-CS for our subsequent analyses. In FinnGen, 61,878 individuals met our inclusion criteria for model derivation for CHD and 69,159 for T2D (see "Methods" for details). The median follow-up time was 10.0 years (interquartile range [IQR] 7.8–10.0) for CHD and 10.0 years (IQR 7.5–10.0) for T2D.

We constructed three sex-specific 10-year risk tools for each disease in FinnGen: (1) Baseline; (2) Genomics-enhanced RIsk Tool (GRIT-CHD and GRIT-T2D) combining PRSs with simple and easily surveyable risk factors; and (3) GRIT requiring measurements for systolic blood pressure (SBP) and lipids (GRIT-CHD+ and GRIT-T2D+). We then estimated the predictive performance of the PRSs and the GRIT scores in the UK Biobank for 10-year incident disease (Fig. 1). The validation datasets for the GRIT scores comprised 242,687 UK Biobank individuals with 4469 incident cases for CHD and 121,113 individuals with 2544 incident cases for T2D who met our inclusion criteria (see "Methods", Table 1). Supplementary Table 2 shows the number of incident and prevalent cases by type of event and data source in the validation cohort. Median follow-up was 10.0 years (IQR 8.6–10.0) for CHD and 10.0 years (IQR 8.3–10.0) for T2D.

We tested the incremental value of each risk factor included in the GRIT scores by adding them individually to a model with age and sex in the final FinnGen derivation datasets. The area under the receiver operating characteristic curve (AUC) increments

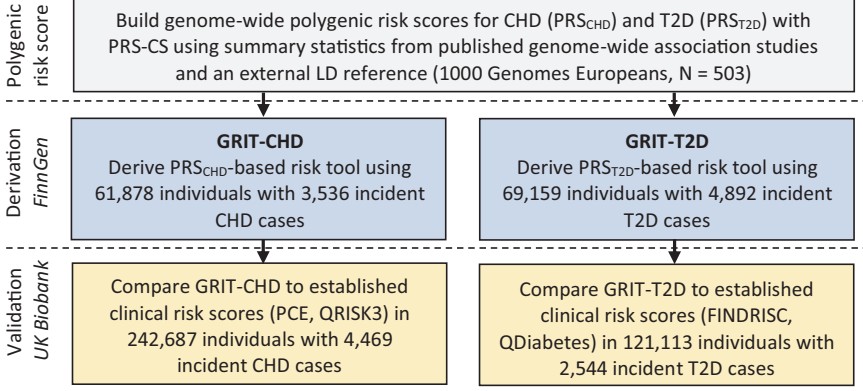

**Fig. 1 Derivation and validation of Genomics-enhanced RIsk Tools (GRIT) for CHD and T2D.** Genome-wide PRSs for CHD and T2D were derived using the algorithm PRS-CS by obtaining weights from GWAS summary statistics from two large GWASs that do not overlap with UK Biobank. We derived two risk tools to estimate 10-year risk of incident disease, GRIT-CHD and GRIT-T2D, which integrate PRSs and simple, easily surveyable risk factors. We then assessed discrimination, reclassification, calibration, and risk stratification of the risk tools in the UK Biobank and compared their performance to established clinical risk scores (Pooled Cohort Equations and QRISK3 for CHD, FINDRISC and QDiabetes for T2D). The derivation and validation of the baseline, GRIT-CHD+, and GRIT-T2D+ scores were analogous.

**Table 1 Baseline participant characteristics.**

| Coronary heart disease | Derivation, N = 61,878 (FinnGen) | | | | Validation, N = 242,687 (UK Biobank) | | | |
|---|---|---|---|---|---|---|---|---|
| | Men, N = 33,774 | | Women, N = 28,104 | | Men, N = 105,439 | | Women, N = 137,248 | |
| | Cases N = 2938 | Non-cases N = 30,836 | Cases N = 598 | Non-cases N = 27,506 | Cases N = 3272 | Non-cases N = 102,167 | Cases N = 1197 | Non-cases N = 136,051 |
| Age, mean ± SD | 60.2 ± 8.1 | 52.7 ± 10.7 | 61.4 ± 8.1 | 51.1 ± 10.7 | 59.9 ± 6.9 | 56.1 ± 8.2 | 61.2 ± 6.4 | 56.4 ± 7.9 |
| Current smoker, n (%) | 1945 (66.2) | 13,978 (45.3) | 148 (24.7) | 5339 (19.4) | 551 (16.8) | 11,816 (11.6) | 216 (18.0) | 11,338 (8.3) |
| Any diabetes[a], n (%) | 233 (7.9) | 1167 (3.8) | 96 (16.0) | 1100 (4.0) | 119 (3.6) | 1804 (1.8) | 30 (2.5) | 1293 (1.0) |
| Use of antihypertensive medications, n (%) | 368 (12.5) | 2933 (9.5) | 196 (32.8) | 3765 (13.7) | 617 (18.9) | 12,021 (11.8) | 261 (21.8) | 14,582 (10.7) |
| BMI, kg/m² , mean ± SD | 27.6 ± 4.1 | 27.0 ± 4.1 | 28.4 ± 5.5 | 26.6 ± 5.2 | 28.1 ± 4.0 | 27.4 ± 4.0 | 27.6 ± 4.9 | 26.7 ± 4.9 |
| Family history of CHD, n (%) | n/a | n/a | n/a | n/a | 1599 (48.9) | 37,375 (36.6) | 672 (56.1) | 59,195 (43.5) |

| Type 2 diabetes | Derivation, N = 69,159 (FinnGen) | | | | Validation, N = 121,113 (UK Biobank) | | | |
|---|---|---|---|---|---|---|---|---|
| | Men, N = 38,861 | | Women, N = 30,298 | | Men, N = 55,898 | | Women, N = 65,215 | |
| | Cases N = 3234 | Non-cases N = 35,627 | Cases N = 1658 | Non-cases N = 28,640 | Cases N = 1532 | Non-cases N = 54,366 | Cases N = 1012 | Non-cases N = 64,203 |
| Age, mean ± SD | 58.9 ± 8.5 | 54.2 ± 10.9 | 57.6 ± 9.7 | 52.1 ± 11.1 | 60.2 ± 7.0 | 57.2 ± 8.1 | 59.9 ± 7.1 | 56.9 ± 7.9 |
| Current smoker, n (%) | 1612 (49.8) | 16,656 (46.8) | 374 (22.6) | 5501 (19.2) | 218 (14.2) | 6076 (11.2) | 136 (13.4) | 5406 (8.4) |
| Use of antihypertensive medications, n (%) | 948 (29.0) | 4644 (13.0) | 646 (39.0) | 4727 (16.5) | 669 (43.7) | 11,365 (20.9) | 366 (36.2) | 9254 (14.4) |
| Statin therapy, n (%) | 458 (14.2) | 2,274 (6.4) | 280 (16.9) | 1,902 (6.6) | 551 (36.0) | 9645 (17.7) | 257 (25.4) | 5795 (9.0) |
| BMI, kg/m² , mean ± SD | 30.3 ± 4.7 | 26.7 ± 3.8 | 31.6 ± 6.0 | 26.4 ± 4.9 | 31.1 ± 4.7 | 27.5 ± 3.9 | 31.8 ± 6.0 | 26.7 ± 4.8 |
| Prevalent CVD, n (%) | 744 (23.0) | 3,896 (11.0) | 217 (13.1) | 1494 (5.2) | 350 (22.8) | 4811 (8.9) | 124 (12.3) | 2430 (3.8) |
| Gestational diabetes, n (%) | n/a | n/a | 64 (3.9) | 473 (1.7) | n/a | n/a | 22 (2.2) | 220 (0.3) |
| Family history of T2D, n (%) | n/a | n/a | n/a | n/a | 452 (29.5) | 9756 (17.9) | 403 (39.8) | 13,268 (20.7) |

The baseline participant characteristics are shown for cases and non-cases in the derivation (FinnGen) and validation (UK Biobank) datasets for CHD and T2D stratified by sex.
Age age at enrollment, BMI body mass index.
[a]T1D, T2D, and unspecified diabetes.

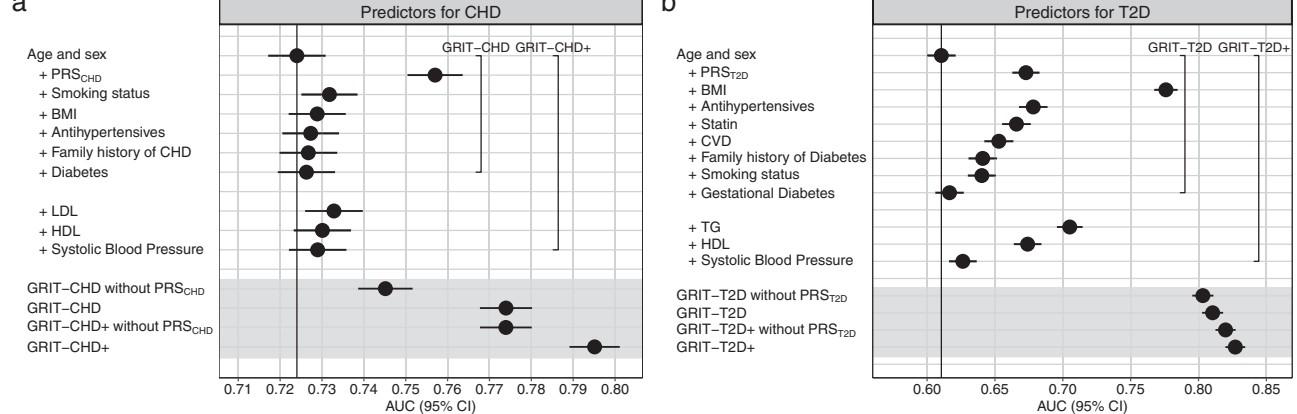

**Fig. 2 AUC for combinations of age, sex, and individual risk factors and the GRIT scores in UK Biobank.** Panel **a** shows results for CHD (N = 242,687 participants) and panel **b** for T2D (N = 121,113). The AUC was first calculated for age and sex and additionally for each individual risk factor integrated with age and sex. Lastly, the AUC was calculated for the GRIT scores and the GRIT scores without PRSs. Points indicate AUC estimates and error bars represent the 95% CIs for each factor with incident disease as endpoint. BMI body mass index, LDL low-density lipoprotein, HDL high-density lipoprotein, TG triglycerides, CVD cardiovascular disease.

ranged from +0.002 to +0.03 in CHD and from +0.006 to +0.17 in T2D in UK Biobank (Fig. 2). The largest increments in AUC over age and sex came from PRS_CHD (in CHD analyses) and BMI (in T2D analyses). The models combining PRSs and the conventional risk factors, GRIT-CHD and GRIT-CHD+, had AUCs of 0.774 (95% confidence interval [CI] 0.768–0.780) and 0.795 (0.789–0.801), respectively. Similarly, GRIT-T2D and GRIT-T2D+ had AUCs of 0.810 (0.803–0.818) and 0.827 (0.820–0.834).

Subgroup analyses displayed better absolute discrimination for women and younger individuals, and non-obese (BMI < 30) individuals in T2D. The greatest performance was seen in individuals younger than 55 years old at baseline, with AUCs of 0.805 (0.791–0.819) for GRIT-CHD and 0.846 (0.830–0.861) for GRIT-T2D (Fig. 3, Supplementary Table 3).

**Discrimination of GRIT versus clinical risk scores.** Next, we compared our risk tools to routinely used clinical risk calculators:

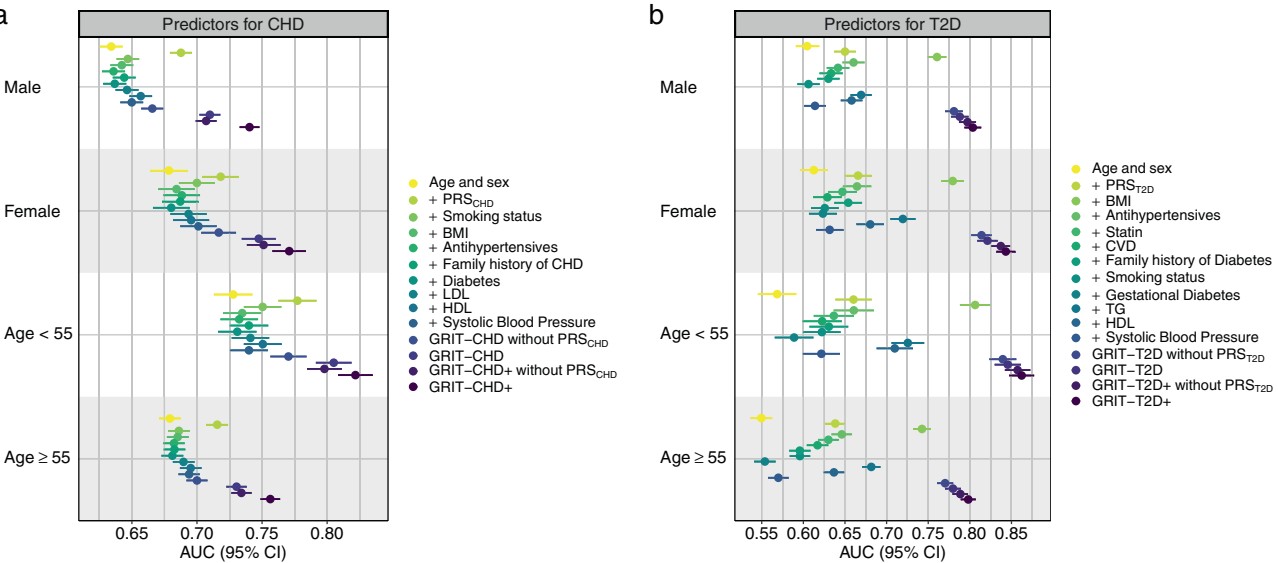

**Fig. 3 AUC for combinations of age, sex, and individual risk factors and the GRIT scores in UK Biobank stratified by sex and age.** Panel **a** shows results for CHD and panel **b** for T2D. The AUC was first calculated for age and sex and additionally for each individual risk factor integrated with age and sex. Lastly, the AUC was calculated for the GRIT scores and the GRIT scores without PRSs. Points indicate AUC estimates and error bars represent the 95% CIs for each factor with incident disease as endpoint. The CHD analysis sample sizes were 105,439 (men), 137,248 (women), 101,508 (age < 55), and 141,179 (age ≥ 55). The T2D analysis sample sizes were 55,898 (men), 65,215 (women), 46,238 (age < 55), and 74,875 (age ≥ 55). BMI body mass index, LDL low-density lipoprotein, HDL high-density lipoprotein, TG triglycerides, CVD cardiovascular disease.

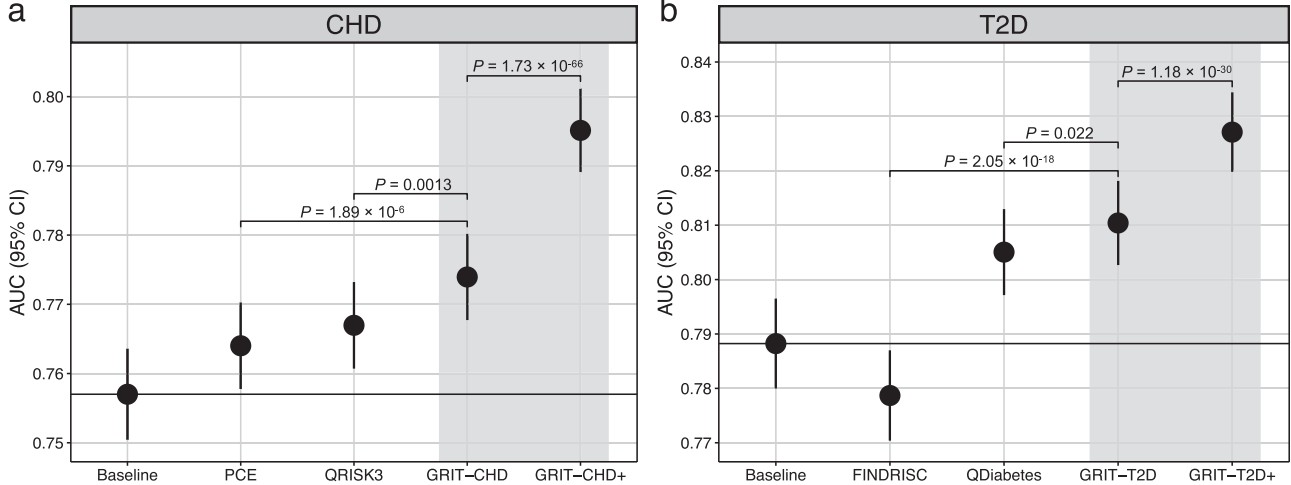

**Fig. 4 AUC for the GRIT scores compared with the clinical risk scores in UK Biobank.** Panel **a** shows results for CHD ($N = 242,687$ participants) and panel **b** for T2D ($N = 121,113$). The sex-specific baseline models include age and PRS, and additionally BMI for the T2D model. Our sex-specific Genomics-enhanced RIsk Tools (GRIT-CHD and GRIT-T2D) were compared to established clinical risk scores (Pooled Cohort Equations and QRISK3 for CHD, FINDRISC and QDiabetes for T2D). Points indicate AUC estimates and error bars represent the 95% CIs for each factor with incident disease as endpoint. All tests were two-sided. PCE Pooled Cohort Equations.

GRIT-CHD to the QRISK3 and Pooled Cohort Equations (PCE) algorithms and the GRIT-T2D to the QDiabetes and the Finnish Diabetes Risk Score (FINDRISC) algorithms (Fig. 4 and Supplementary Fig. 1). We used sex-specific baseline models that included age and PRS, and additionally BMI for the T2D model analyses, as benchmarks for the performance of both the clinical risk scores and the GRIT scores. Measured with AUC, the discrimination of the baseline model for CHD was 0.756 (0.750–0.763) and for the baseline model for T2D, 0.789 (0.780–0.797). GRIT-CHD had a slightly higher AUC compared to QRISK3 (incremental AUC +0.007 against QRISK3, $P = 0.0013$), and ranging from -0.006 to 0.008 in sex and age

subgroups. The AUC increment over PCE was slightly larger (+0.010, $P = 1.89 \times 10^{-6}$) and ranged from 0.003 to 0.015 in the subgroups. For the GRIT-T2D, the AUC compared to QDiabetes was similar (incremental AUC +0.005, $P = 0.022$) and from 0.003 to 0.011 in sex, age, and BMI subgroups. Lastly, the AUC increment over FINDRISC was larger (+0.031, $P = 2.05 \times 10^{-18}$), ranging from 0.022 to 0.049 in subgroup analysis. Enriching GRIT-CHD and GRIT-T2D by clinical measurements (GRIT-CHD+ and GRIT-T2D+) resulted in pronounced and statistically significant ($P < 2.2 \times 10^{-16}$ for all comparisons) performance improvements for the GRIT-CHD+ (incremental AUC +0.028 against QRISK3 and +0.031 against PCE) and the

**Table 2 NRI between GRIT scores and clinical risk scores in UK Biobank.**

| | | Upclassified to higher risk (%) | Both high risk (%) | Downclassified to lower risk (%) | Both low risk (%) | Category-based NRI [95% CI] |
|---|---|---|---|---|---|---|
| GRIT-CHD vs PCE | Cases | 428 (9.6%) | 320 (7.2%) | 221 (4.9%) | 3500 (78.3%) | 4.7 [3.6, 6.0] |
| | Non-cases | 5463 (2.3%) | 3110 (1.3%) | 3864 (1.6%) | 225,781 (94.8%) | −0.7 [−0.8, −0.6] |
| | All | 5891 (2.4%) | 3430 (1.4%) | 4085 (1.7%) | 229,281 (94.5%) | 4.0 [3.0, 5.3] |
| GRIT-CHD+ vs PCE | Cases | 616 (13.8%) | 411 (9.2%) | 130 (2.9%) | 3312 (74.1%) | 10.9 [9.7, 12.0] |
| | Non-cases | 6903 (2.9%) | 4356 (1.8%) | 2618 (1.1%) | 224,341 (94.2%) | −1.8 [−1.9, −1.7] |
| | All | 7519 (3.1%) | 4767 (2.0%) | 2748 (1.1%) | 227,653 (93.8%) | 9.1 [7.9, 10.2] |
| GRIT-T2D vs QDiabetes | Cases | 330 (13.0%) | 628 (24.7%) | 108 (4.2%) | 1478 (58.1%) | 8.7 [7.1, 10.4] |
| | Non-cases | 4453 (3.8%) | 4603 (3.9%) | 1452 (1.2%) | 108,061 (91.1%) | −2.5 [−2.6, −2.4] |
| | All | 4783 (3.9%) | 5231 (4.3%) | 1560 (1.3%) | 109,539 (90.4%) | 6.2 [4.6, 7.8] |
| GRIT-T2D+ vs QDiabetes | Cases | 463 (18.2%) | 654 (25.7%) | 82 (3.2%) | 1345 (52.9%) | 15.0 [13.1, 16.7] |
| | Non-cases | 5840 (4.9%) | 4727 (4.0%) | 1328 (1.1%) | 106,674 (90.0%) | −3.8 [−3.9, −3.7] |
| | All | 6303 (5.2%) | 5381 (4.4%) | 1410 (1.2%) | 108,019 (89.2%) | 11.1 [9.3, 12.9] |

The reclassification numbers and proportion of cases, non-cases, and all participants were assessed alongside the category-based net reclassification improvement (event NRI, nonevent NRI, overall NRI) by comparing the Genomics-enhanced RIsk Tools (GRIT-CHD and GRIT-CHD+ for CHD, GRIT-T2D and GRIT-T2D+ for T2D) to established clinical risk scores (PCE for CHD and QDiabetes for T2D). Based on the established clinical risk scores, the thresholds for 10-year risk were 7.5% for CHD and 5.6% for T2D.

GRIT-T2D+ (+0.022 against QDiabetes and +0.048 against FINDRISC).

**Calibration and reclassification**. To assess model calibration in UK Biobank, we plotted the observed incidences across deciles of predicted risk. All risk models, including the established clinical risk calculators, showed poor calibration by overestimating absolute risk. After recalibration by estimating the baseline hazard and mean component in the validation datasets, we observed greatly improved goodness-of-fit (Supplementary Figs. 2 and 3) in all but the FINDRISC model which we did not recalibrate and therefore did not consider in downstream analyses. Model discrimination was similar after recalibration (Supplementary Fig. 4).

Next, we used the recalibrated estimates to assess reclassification over risk thresholds broadly applied for the established risk calculators (10-year risk ≥7.5% for CHD and ≥5.6% for T2D) (Table 2). We compared the GRIT-CHD and GRIT-CHD+ to PCE and the GRIT-T2D and GRIT-T2D+ to QDiabetes. The event net reclassification improvement (NRI), interpretable as the net percentage of individuals with the event of interest correctly upclassified, was 4.7 (GRIT-CHD), 10.9 (GRIT-CHD+), 8.7 (GRIT-T2D), and 15.0 (GRIT-T2D+) in incident disease cases. In non-cases, the nonevent NRIs (the net percentage of individuals without the event of interest correctly downclassified) were −0.7 (GRIT-CHD), −1.8 (GRIT-CHD+), −2.5 (GRIT-T2D) and −3.8 (GRIT-T2D+).

Overall categorical NRI (the sum of the proportion of cases that are reclassified to a higher risk category and the proportion of non-cases reclassified to a lower category) was 4.0 (95% CI 3.0–5.3) for the GRIT-CHD, 9.1 (7.9–10.2) for the GRIT-CHD+, 6.2 (4.6–7.8) for the GRIT-T2D and 11.1 (9.3–12.9) for the GRIT-T2D+. The continuous NRI and the integrated discrimination improvement (IDI) had positive and statistically significant shifts, with the exception for GRIT-T2D against QDiabetes (Supplementary Table 4). The reclassification of individuals, NRI, and IDI against QRISK3 is detailed in Supplementary Tables 5 and 6.

**Risk stratification**. Lastly, we estimated the effects across risk categories of GRIT-CHD and GRIT-T2D, comparing individuals between the 25 and 75th percentiles and the top 5% of the distribution. For GRIT-CHD, the hazard ratio (HR) for the top 5% was 6.23 (95% CI 5.73–6.77) and for GRIT-T2D, 10.9 (9.84–12.2). Similarly, for GRIT-CHD+ the HR for the top 5% was 8.28 (7.62–9.00) and for GRIT-T2D+ 12.6 (11.3–14.0). The

corresponding HRs for the clinical risk scores were 5.43 (5.00–5.91) for QRISK3, 5.54 (5.10–6.02) for PCE, 10.3 (9.31–11.5) for QDiabetes, and 6.83 (6.14–7.60) for FINDRISC. The Kaplan–Meier estimates of cumulative incidence for the GRIT scores are shown in Supplementary Fig. 5. The cross-predictive performance of the GRIT scores is shown in Supplementary Fig. 6.

## Discussion

In this cross-biobank analysis utilizing longitudinal health and genomics data from Finland and the United Kingdom, we derived Genomics-enhanced RIsk Tools, GRIT-CHD, and GRIT-T2D, to serve as early risk indicators for two common cardiometabolic diseases, CHD and T2D. First, we show that PRSs for CHD and T2D greatly benefit from being used in risk prediction together with simple, easily surveyable risk factors. Secondly, we show that GRIT-CHD and GRIT-T2D, which integrate PRSs with risk factors easily obtainable by questionnaires, have at least comparable performance to established risk scores recommended in respective clinical practice guidelines. Third, we demonstrate that further enriching the GRIT scores with routine clinical measurements results in further performance improvements, improving prediction beyond established clinical risk scores. Together these findings show that PRSs combined with simple clinical risk factors could serve in early risk identification of individuals in need of further clinical risk assessment and targeted prevention.

PRSs have been shown to perform well in predicting lifetime risk, demonstrating usefulness in targeted identification of high-risk individuals missed by traditional methods for risk stratification[4,5]. In previous studies investigating the incremental value of PRSs integrated to clinical risk scores, CHD PRSs have yielded varying performance improvements over routinely used clinical risk assessment tools, such as PCE or QRISK3[4,16–18]. For T2D, the improvements with PRS over clinical risk assessment tools have been moderate[4,19]. Here, when added on top of with age and sex, the $PRS_{CHD}$ had a higher AUC for incident CHD than any of the eight other individual risk factors of GRIT-CHD. A largely corresponding effect of the $PRS_{CHD}$ was recently demonstrated also among symptomatic patients with suspected CHD[20]. Similarly, when added on top of with age and sex, the $PRS_{T2D}$ had a similar or a higher AUC for incident T2D than seven of the ten included conventional risk factors.

Our GRIT-CHD and GRIT-T2D scores showed at least comparable predictive performance with widely applied clinical risk

scores for CHD and T2D, and the overall reclassification improvements with GRIT-CHD and GRIT-T2D were driven by improvements particularly in incident disease cases. In addition to categorical NRI, the continuous NRI and IDI, metrics not based on fixed risk thresholds, showed a similar trend in improved reclassification over clinical risk scores. The GRIT scores upclassified many non-cases to a higher risk at the thresholds aligned with the established clinical risk scores, but considering that they were derived to serve as preliminary risk indicators, the harms caused by these false-positive classifications are minimal as the individuals would still require more detailed clinical risk assessment before possible preventative interventions. In addition to $PRS_{CHD}$, the GRIT scores for CHD include age, sex, smoking status, BMI, blood-pressure-lowering medication use, history of diabetes, and family history of CHD and additionally SBP, high-density lipoprotein (HDL), and low-density lipoprotein (LDL) in GRIT-CHD+, all of which are also included in QRISK3 and mostly in PCE (both algorithms use total cholesterol [TC] instead of LDL). Similarly, in addition to $PRS_{T2D}$, the GRIT scores for T2D include age, sex, BMI, smoking status, current blood-pressure-lowering medication use, current statin use, history of cardiovascular disease (CVD), history of gestational diabetes, and family history of diabetes and additionally SBP, triglycerides (TG), and HDL in GRIT-T2D+, all of which (except SBP and lipid measurements) are also included in QDiabetes and most of which also in FINDRISC. The clinical risk scores also include components we could not include in the GRIT scores due to data limitations, such as measures for diet, physical activity, socioeconomic factors, and waist circumference, but their effects may be mediated to an extent by risk factors that were included, such as smoking and BMI. Therefore, in addition to PRSs, the higher predictive value of the GRIT scores compared to the clinical risk scores in UK Biobank is also likely to be impacted by differences in the model input variables, model complexity, and participant characteristics between the derivation cohorts.

In contrast to previous studies, we studied the role of PRSs combined with clinical risk factors obtainable by simple questionnaires, without the need of additional laboratory and clinical measurements such as blood pressure and lipids. With the decreasing costs of sequencing, rapidly expanding knowledge of genomics, and increasing public interest, genome data is becoming increasingly available for applications of disease prevention and care[1,2]. Our approach allows for efficient and relatively effortless risk estimation across resources where genome data is readily available, but measurements of quantitative risk factors used in clinical care such as lipids might not be. Examples include population-based biobanks and commercial genomics databases, or when clinical variables from healthy, asymptomatic individuals have not yet been measured. As the effects of unfavorable lifestyle and high genetic risk accumulate over the course of life, early identification of high-risk individuals is critical to effectively manage cardiometabolic disease risk. Moreover, individuals with a high PRS are more likely to benefit from preventative efforts for CHD and T2D, such as healthy lifestyle changes[21,22] and statin therapy[23,24].

While we observed improved performance in all included subgroups when adding supplemental risk factors to PRS-based risk models, the PRSs and the GRIT scores had the best discrimination in women and younger individuals, highlighting the performance of PRS-based risk tools in these groups of individuals in which clinical risk scores often have limited utility[7]. As both the GRIT scores and the clinical risk scores failed to classify as high risk many of the individuals who had first disease events during the follow-up, continuous preventative efforts are crucial to combat the development of both lifestyle and clinical risk factors early on in younger individuals, particularly in those at

high polygenic risk, to reduce the socioeconomic impact of cardiometabolic diseases on society.

The study has several strengths. It uses extensive health registry data with high validity from both Finland and the United Kingdom[25,26]. Compared to most previous studies in the UK Biobank, we used a recent release of incident data with a median follow-up of over 10 years and used an interim primary care data release from general practices across the United Kingdom. This enabled more accurate identification of conditions often diagnosed outside of a hospital inpatient setting, such as T2D, where nearly half of incident cases are recorded only in primary care databases[27]. The impact of utilizing primary care data in this study should be relatively small, as the number of CHD and T2D cases identified using only the general practice data was small. However, the primary care data allowed us to calculate the QRISK3 and QDiabetes clinical scores more accurately as they were derived using the same data sources[28,29]. Following the recommendations in recently published reporting guidelines for genetic prediction studies[30], and in contrast to previous studies[5,16–19], we used the external UK Biobank cohort for out-of-sample evaluation of our risk tools. As the predictive performance of risk models can greatly worsen when assessed outside of their derivation datasets[31], our results indicate that GRIT-CHD and GRIT-T2D may generalize well also to other cohorts.

The study needs to be interpreted considering some limitations. The QRISK3 and PCE algorithms were optimized to predict 10-year risk of CVD rather than CHD alone, which may affect their performance. We did not consider rare genetic variants known to considerably increase disease risk in some individuals, such as in familial hyperlipidemia[32]. Additionally, our data was limited to middle-aged individuals of European ancestry, which increases the risk of lack of generalizability outside of European populations. Expanding validation of the GRIT-CHD and GRIT-T2D scores to more diverse populations is needed before possible clinical implementation, including integration of ancestry-specific PRSs to improve applicability across ethnic groups. Despite a very low response rate resulting in oversampling of healthy individuals in UK Biobank, the risk factor associations have been considered to be generalizable[33,34]. As data on some clinical risk factors were not available in FinnGen, we used effect sizes from our previous analyses of FinnGen subcohorts with more phenotypic data available. Finally, as a small number of overlapping input weight samples for the $PRS_{CHD}$ within the FinnGen study possibly inflated the effect size of polygenic risk in the joint modeling, the performance of the GRIT-CHD scores in UK Biobank may have been decreased to some degree.

In conclusion, our approach demonstrates an effective application of genome data in a prospective setting, offering individuals accessible and simple, yet powerful disease risk assessment without the need for blood tests and initial evaluation in a healthcare setting. In addition to CHD and T2D, similar PRS-based approaches as described here could also be useful for improving risk prediction in a number of other complex diseases, such as common cancers and atrial fibrillation, by targeting more detailed risk assessment and earlier preventative interventions to high-risk individuals. Further research is warranted to explore the practical application and effectiveness of genome-based risk information in the clinical setting.

## Methods

**Study populations.** FinnGen Data Freeze 7 with 309,154 Finnish individuals comprises prospective epidemiological and disease-based cohorts and hospital biobank samples. The data were linked by the unique national personal identification numbers to national hospital discharge (available from 1968), death (1964–), cancer (1953–) and medication reimbursement (1964–) and purchase (1995–) registries. To have sufficient follow-up and representative estimates in our analyses, we selected individuals aged 30 to 75 at recruitment who were recruited before

2017 (N = 95,182). Of these individuals, we excluded 20,413 (21.2% of the dataset) with no recorded BMI or smoking status, and additionally 10,471 (11.0%) with prevalent CVD and 8839 (9.3%) with statin use at baseline (in CHD analyses), and 7877 (8.3%) with prevalent diabetes (in T2D analyses). After these exclusions, the derivation datasets included 61,878 individuals with 3536 incident cases for CHD, and 69,159 with 4892 incident cases for T2D (Supplementary Table 7). The baseline characteristics were similar between the included and excluded individuals (Supplementary Tables 8 and 9).

The UK Biobank is a prospective cohort study that recruited over 500,000 participants from the United Kingdom between 2006 and 2010[25]. Age at baseline ranged from 40 to 69. At baseline, individuals completed extensive questionnaires and an interview by a trained nurse about sociodemographic, lifestyle, and health-related factors, along with a range of physical and biomarker measurements. The British ancestry subset included 343,672 unrelated individuals with 18,698 cases of CHD and 24,192 cases of T2D. For our CHD analyses, we excluded 24,133 individuals (7.0% of the dataset) with prevalent CVD and further 38,632 individuals (11.2%) using statins at baseline. We additionally excluded 38,220 individuals (11.1%) in the CHD analyses with missing data on predictors required to calculate our risk tools and clinical risk scores. The final CHD validation dataset comprised 242,687 individuals with 4469 incident CHD cases.

A subset of the British ancestry dataset (N = 160,338) had primary care data available and was chosen for T2D analyses. Further 7722 individuals (4.8% of individuals with primary care data) with prevalent diabetes and 31,503 individuals (19.6%) who had missing data on predictors were excluded. The final T2D validation dataset included 121,113 individuals with 2544 incident T2D cases. We ascertained disease outcomes based on linkage to hospital inpatient episodes, the death registry from the Office of National Statistics and the primary care data from English, Scottish, and Welsh GP practices provided to the UK Biobank from four different data providers. The baseline characteristics were similar between the included and excluded individuals in UK Biobank (Supplementary Tables 10 and 11).

**Disease endpoints and risk factor definitions**. Detailed descriptions of endpoints and risk factors are described in Supplementary Data 1 and 2. We defined disease outcomes and morbidities in FinnGen following the International Classification of Diseases (ICD) ICD-10, ICD-9, and ICD-8, Nordic Medico-Statistical Committee (NOMESCO), and National League of Hospitals classifications, and Finnish Heart Patients V1 and V2 codes. Similarly, in UK Biobank, we used ICD-10 and ICD-9 codes together with OPCS Classification of Interventions and Procedures version 4 (OPCS-4) and General Practice Read V2 and Read V3 clinical codes to identify comorbidities and cases for our main disease outcomes. To capture prevalent morbidities at baseline, we used also non-cancer illness codes (UK Biobank field 20002) recorded by healthcare professionals at the UK Biobank assessment centers. To capture prevalent cases of diabetes from the UK Biobank baseline data for exclusion and as covariates to our analyses, we used methods proposed by East-wood et al. to identify probable and possible (i.e., likely) prevalent cases of type 1 diabetes (T1D), T2D, and gestational diabetes[27].

Primary care clinical codes and self-reported prescription medication codes (field 20003) were identified using the UK Biobank coding system lookups and mapping[35]. We identified primary care clinical codes (Read V2 and Read V3) used in recording primary care data using corresponding ICD-10 codes and examined matched codes by hand to detect anomalies. We manually matched relevant codes for regularly taken prescription medications recorded by a trained nurse (field 20003) following the Anatomical Therapeutic Chemical Classification System (ATC) and the British National Formulary (BNF).

Follow-up ended at first diagnosis of the disease of interest, death, or at the censoring date December 31, 2019 (FinnGen), or at the censoring date of hospital inpatient data (UK Biobank; English hospital inpatient records up to May 2020, Scottish up to November 2016, Welsh up to March 2016), whichever came first.

**Genotyping, imputation, and PRS calculation**. FinnGen individuals were geno-typed with Illumina and Affymetrix arrays (Illumina Inc., San Diego, CA, USA, and Thermo Fisher Scientific, Santa Clara, CA, USA). Genotype calls were made with the GenCall or zCall (for Illumina) and the AxiomGT1 algorithm (for Affymetrix). Genotype imputation was performed with Beagle 4.1 (described in the following protocol: https://doi.org/10.17504/protocols.io.xbgfijw) by using the SISu v3 population-specific reference panel developed from high-quality data for 3775 high-coverage (25–30×) whole-genome sequences in Finns. Samples with ambiguous gender, high genotype missingness (>5%), excess heterozygosity (+-4SD) and non-Finnish ancestry were excluded, as well as all variants with high missingness (>2%), low Hardy–Weinberg equilibrium $p$ value ($< 1 \times 10^{-6}$), and minor allele count (MAC < 3). BCFtools 1.7 and 1.9 and PLINK v2.00a2.3LM were used for data and variant handling and PRS calculation. Cromwell 61 was used for workflow handling. Array data pre-phasing was carried out with Eagle 2.3.5[36] with the number of conditioning haplotypes set to 20,000.

UK Biobank participants were genotyped using the Affymetrix UK BiLEVE Axiom array[37] or the Affymetrix UK Biobank Axiom array[25]. The dataset has been phased and imputed centrally using the Haplotype Reference Consortium and the merged UK10K and 1000 Genomes (phase 3) reference panels[38,39]. We limited our

analyses to 343,672 unrelated British ancestry individuals passing genotype imputation quality control.

We built genome-wide PRSs for CHD ($PRS_{CHD}$) and T2D ($PRS_{T2D}$) with the software PRS-CS[40] (PRS-CS-auto, with 1000 Genomes Project European sample, N = 503, as the external linkage disequilibrium [LD] reference panel) using HapMap3 variants. The input weights came from two large GWASs independent of UK Biobank, but with some sample overlap in FinnGen[12,13], as we could not accurately exclude the overlapping samples for CHD (1208 individuals free of CHD in the FINRISK 1992–2007 cohorts included in FinnGen) from the FinnGen dataset. Our PRSs showed acceptable goodness-of-fit. The final variant counts for the PRS-CS PRSs were 1,090,048 for CHD and 1,091,673 for T2D in FinnGen and 1,087,714 for CHD and 1,089,342 for T2D in UK Biobank.

**Derivation of risk tools in FinnGen**. We derived three sex-specific 10-year risk tools for both CHD and T2D in FinnGen. We used only incident cases and excluded individuals who had prevalent CVD or used statins at baseline (in CHD analyses) or individuals with prevalent diabetes (in T2D analyses) and individuals with missing data on predictors. To derive the risk tools, we used a Cox proportional hazards model to estimate beta coefficients, baseline hazard, and mean component, adjusting for the first 10 principal components of Finnish ancestry and stratified the analyses by sex. The tools (linear predictors from Cox proportional hazards models) were based on risk factors available in the FinnGen datasets ($PRS_{CHD}$ or $PRS_{T2D}$, sex, age, smoking status, BMI, blood-pressure-lowering medication use, statin use, history of diabetes, gestational diabetes and CVD), and we additionally integrated beta coefficients for self-reported first-degree family history and clinical measurements (SBP, HDL, LDL, and TG) from FinnGen subcohorts[4,41] to the linear predictors derived in FinnGen. The beta coefficients for family history and clinical measurements were from our previous analyses from multivariate models (including PRS) for incident CHD/T2D obtained from the population-based cohort FINRISK[4,41], which is included in FinnGen.

The three sex-specific risk tools for incident CHD included (1) $PRS_{CHD}$ and age; (2) $PRS_{CHD}$, age, current smoking status, BMI, current blood-pressure-lowering medication use, history of diabetes, and self-reported first-degree family history of CHD; and (3) Model 2 integrated with clinical measurements for SBP, HDL, and LDL. For incident T2D, the three sex-specific risk tools included (1) $PRS_{T2D}$, age, and BMI; (2) $PRS_{T2D}$, age, BMI, current smoking status, current blood-pressure-lowering medication use, current statin use, history of CVD, history of gestational diabetes (women only), and self-reported first-degree family history of diabetes; and (3) Model 2 integrated with clinical measurements for SBP, HDL, and TG. We separately evaluated the incremental value of each individual risk factor when added on top of with age and sex. The beta coefficients, baseline hazard, and mean component of all risk models are detailed in Supplementary Tables 12–15.

Criteria for inclusion of individual risk factors to our risk models were consistency of the association in established literature, data availability in FinnGen, and being an independent risk factor in sex-specific multivariate Cox proportional hazards models with $P < 0.01$.

**Risk tool validation in UK Biobank**. The risk tools derived in FinnGen were tested in UK Biobank first by using the models' original baseline hazard and mean component estimated in FinnGen and second by estimating and integrating the baseline hazard and mean component from the UK Biobank validation datasets to the linear predictor alongside the original beta coefficients following a recalibration process, as detailed below. For both CHD and T2D, we compared our tools to two established clinical risk scores measuring 10-year risk.

For CHD, we compared our risk tools to two algorithms for CVD (a composite outcome including CHD) prevention: the QRISK3 risk algorithm and the American College of Cardiology/the American Heart Association 2013 PCE. QRISK3 is the latest version of the QRISK algorithm and is currently recommended by the National Institute for Health and Care Excellence for use in the United Kingdom for disease prevention in primary care, with 10% absolute risk as threshold for recommending preventative treatments such as statins[28]. The PCE is a US-derived risk score with preventative statin treatment recommended in individuals with an elevated risk (absolute risk over 7.5%)[8]. Both QRISK3 and PCE are sex-specific and include age, TC, HDL, SBP, blood-pressure-lowering medication, diabetes, and smoking status as covariates. QRISK3 also includes BMI, additional comorbidities, as well as socioeconomic risk factors.

For T2D, we compared our risk tools to the QDiabetes-2018 score and the FINDRISC algorithm. The QDiabetes-2018 score is the latest version of the QDiabetes algorithm validated for use in the United Kingdom population, with a risk threshold of 5.6% absolute 10-year risk selected to optimize sensitivity for identifying individuals for further risk evaluation[29]. FINDRISC is a widely utilized count-based risk tool originally derived in a Finnish population-based sample. FINDRISC has been validated in numerous studies and populations, with an optimal risk threshold in the range of 11–14 points[42]. Both tools include age, sex, BMI, use of blood-pressure-lowering medication or diagnosis of hypertension, history of high blood glucose or gestational diabetes, and family history of any diabetes as covariates. QDiabetes includes additional comorbidities and socioeconomic risk factors, and we used a nonlaboratory model (Model A) that does not include diagnostic blood tests for T2D (fasting glucose or HbA1c). FINDRISC additionally considers waist circumference and measures of physical

activity as well as consumption of vegetables, fruits, and berries as risk factors. Current smokers with missing data on smoking frequency were set as moderate smokers in the QRISK3 and QDiabetes algorithms.

Calculating risks with the original clinical risk calculators required a few small modifications in UK Biobank: first, the QRISK3 and QDiabetes algorithms consider use of drugs at baseline as at least two prescriptions, with the most recent one no later than 28 days before the date of cohort entry[28,29]. The UKB treatment/medication codes (field 20003) include regularly taken medications at baseline. Second, we used the reported means from the QRISK3 derivation and validation cohorts for SBP variability, 9.3 (women) and 9.9 (men), as this data was not available in UKB. Third, in QRISK3, family history of disease is defined as disease cases of CHD aged less than 60 years, but in UK Biobank, fields 20107 (illnesses of father), 20110 (illnesses of mother), and 20111 (illnesses of siblings) contain only family history of heart disease without specifications. Fourth, the FINDRISC algorithm assigns values of five and three for positive history in first- and second-degree relatives, respectively[42]. Second-degree family history was not available in UK Biobank. UK Biobank datafields used to calculate the clinical risk scores are described in Supplementary Data 3.

**Statistics and reproducibility**. We restricted the follow-up for incident disease analyses to a maximum of ten years after baseline. We used only complete cases with respect to the diseases and predictors, with the exception for smoking frequency in UK Biobank. We used Schoenfeld residuals and log–log inspection to assess the proportional hazards assumption.

In external validation in UK Biobank, we assessed metrics for discrimination, calibration, goodness-of-fit, and reclassification. To assess discrimination of our risk models and clinical risk scores, we computed the AUC with 95% CIs. AUC estimates with 95% CIs (nonparametric approach, DeLong) and AUC comparisons were calculated with the R package pROC. We also assessed AUC separately in men and women, stratifying by age and BMI (in T2D analyses), with a cut-off at 55 years in both the CHD and the T2D analyses and a BMI of 30 in T2D analyses. Calibration and goodness-of-fit were assessed graphically by plotting the absolute risks against the mean predicted probability within deciles of the predicted probabilities. We recalibrated all models except FINDRISC by using Cox proportional hazards models to estimate the baseline survival function and calculated the mean component (the sum of the predictor variable means weighted by respective coefficients) in the final UK Biobank validation datasets and combined these with the models' published coefficients in the risk equations to obtain recalibrated 10-year risk estimates (recalibration parameters in Supplementary Table 16). We calculated categorical NRI[43] values for the recalibrated models over established clinical risk thresholds and obtained bootstrapped 95% CIs based on 200 replications. We also calculated the continuous NRI and IDI[44] values after recalibration. A Cox proportional hazards model was used to estimate HRs and 95% CIs for the risk models. Kaplan–Meier survival curves were estimated using the R package survminer. All statistical tests were two-sided. Analyses were performed in R versions 4.1.1 (FinnGen) and 3.6.0 (UK Biobank).

**Ethics statement**. Patients and control subjects in FinnGen provided informed consent for biobank research, based on the Finnish Biobank Act. Alternatively, separate research cohorts, collected prior the Finnish Biobank Act came into effect (in September 2013) and start of FinnGen (August 2017), were collected based on study-specific consents and later transferred to the Finnish biobanks after approval by Fimea (Finnish Medicines Agency), the National Supervisory Authority for Welfare and Health. Recruitment protocols followed the biobank protocols approved by Fimea. The Coordinating Ethics Committee of the Hospital District of Helsinki and Uusimaa (HUS) statement number for the FinnGen study is Nr HUS/990/2017.

The FinnGen study is approved by Finnish Institute for Health and Welfare (permit numbers: THL/2031/6.02.00/2017, THL/1101/5.05.00/2017, THL/341/6.02.00/2018, THL/2222/6.02.00/2018, THL/283/6.02.00/2019, THL/1721/5.05.00/2019, THL/1524/5.05.00/2020, and THL/2364/14.02/2020), Digital and population data service agency (permit numbers: VRK43431/2017-3, VRK/6909/2018-3, VRK/4415/2019-3), the Social Insurance Institution (permit numbers: KELA 58/522/2017, KELA 131/522/2018, KELA 70/522/2019, KELA 98/522/2019, KELA 138/522/2019, KELA 2/522/2020, KELA 16/522/2020, Findata THL/2364/14.02/2020 and Statistics Finland (permit numbers: TK-53-1041-17 and TK/143/07.03.00/2020 (earlier TK-53-90-20).

The Biobank Access Decisions for FinnGen samples and data utilized in FinnGen Data Freeze 7 include: THL Biobank BB2017_55, BB2017_111, BB2018_19, BB_2018_34, BB_2018_67, BB2018_71, BB2019_7, BB2019_8, BB2019_26, BB2020_1, Finnish Red Cross Blood Service Biobank 7.12.2017, Helsinki Biobank HUS/359/2017, Auria Biobank AB17-5154 and amendment #1 (August 17, 2020), Biobank Borealis of Northern Finland_2017_1013, Biobank of Eastern Finland 1186/2018 and amendment 22 § /2020, Finnish Clinical Biobank Tampere MH0004 and amendments (21.02.2020 & 06.10.2020), Central Finland Biobank 1-2017, and Terveystalo Biobank STB 2018001.

This project was conducted with permission of the UK Biobank Resource under application no. 22627. Informed consent was obtained from all UK Biobank participants.

**Reporting summary**. Further information on research design is available in the Nature Research Reporting Summary linked to this article.

## Data availability

The Finnish biobank data can be accessed through the Fingenious® services (web link: https://site.fingenious.fi/en/, email: contact@finbb.fi) managed by FINBB. The UK Biobank resource is available to bona fide researchers for health-related research in the public interest at https://www.ukbiobank.ac.uk/researchers/. The GWAS summary statistics used for constructing our PRSs are available at http://www.cardiogramplusc4d.org/data-downloads/ and https://diagram-consortium.org/downloads.html. LD reference panels constructed using the 1000 Genomes Project[38] phase 3 samples can be downloaded at https://github.com/getian107/PRScs. The weights for our PRSs are available at PGS Catalog[45] (pgs-info@ebi.ac.uk) at https://www.pgscatalog.org/score/PGS001780/ and https://www.pgscatalog.org/score/PGS001781/. Supplementary Data 4 contains the raw data underlying the figures in the main text and Supplementary Information.

## Code availability

The full genotyping and imputation protocol for FinnGen is described at https://doi.org/10.17504/protocols.io.xbgfijw. The PRS-CS pipeline in FinnGen is described in Supplementary Note 1. All software packages and programs used to perform these analyses are freely available, and can be found within the manuscript, Supplementary Information, Supplementary Data, and the Reporting Summary. The code used for these analyses are available from the corresponding author upon reasonable request.

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

## Acknowledgements

We would like to thank Sari Kivikko, Huei-Yi Shen and Ulla Tuomainen for management assistance. The FinnGen project is funded by two grants from Business Finland (HUS 4685/31/2016 and UH 4386/31/2016) and the following industry partners: AbbVie Inc., Astra-Zeneca UK Ltd, Biogen MA Inc., Bristol Myers Squibb (and Celgene Corporation & Celgene International II Sàrl), Genentech., Merck Sharp & Dohme Corp, Pfizer Inc., GlaxoSmithKline Intellectual Property Development Ltd., Sanofi US Services Inc., Maze Therapeutics Inc., Janssen Biotech Inc and Novartis AG. Following biobanks are acknowledged for delivering biobank samples to FinnGen: Auria Biobank (www.auria.fi/biopankki), THL Biobank (www.thl.fi/biobank), Helsinki Biobank (www.helsinginbiopankki.fi), Biobank Borealis of Northern Finland (https://www.ppshp.fi/Tutkimus-ja-opetus/Biopankki/Pages/Biobank-Borealis-briefly-in-English.aspx), Finnish Clinical Biobank Tampere (www.tays.fi/en-US/Research_and_development/Finnish_Clinical_Biobank_Tampere), Biobank of Eastern Finland (www.ita-suomenbiopankki.fi/en), Central Finland Biobank (www.ksshp.fi/fi-FI/Potilaalle/Biopankki), Finnish Red Cross Blood Service Biobank (www.veripalvelu.fi/verenluovutus/biopankkitoiminta) and Terveystalo Biobank (www.terveystalo.com/fi/Yritystietoa/Terveystalo-Biopankki/Biopankki/). All Finnish Biobanks are members of BBMRI.fi infrastructure (www.bbmri.fi). Finnish Biobank Cooperative -FINBB (https://finbb.fi/) is the coordinator of BBMRI-ERIC operations in Finland. The Finnish biobank data can be accessed through the Fingenious® services (https://site.fingenious.fi/en/) managed by FINBB. This research has been conducted using data from UK Biobank, a major biomedical database. This work was supported by the Sigrid Jusélius Foundation (to S.R., M.P.); University of Helsinki HiLIFE Fellow grants 2017–2020 (to S.R.); Academy of Finland Center of Excellence in Complex Disease Genetics (grant number 312062 to S.R., 312076 to M.P.); Academy of Finland (grant number 331671 to N.M., grant number 285380 to S.R., 288509 to M.P.); The Finnish Innovation Fund Tekes (grant number 2273/31/2017 to E.W.); The European Union's Horizon 2020 research and innovation program under grant agreement No 101016775.

## Author contributions

S.R. and N.M. conceived and designed the study. M.T. and N.M carried out the statistical and computational analyses with advice from S.R. and E.W. M.P. coordinated data availability for UK Biobank. The manuscript was written and revised by M.T, N.M., and S.R., with comments from all of the co-authors. All co-authors have approved the final version of the manuscript.

## Competing interests

The authors declare no competing interests.

## Additional information

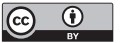

**FinnGen**

**Steering Committee** Aarno Palotie[1] & Mark Daly[1]

**Pharmaceutical companies** Bridget Riley-Gills[5], Howard Jacob[5], Dirk Paul[6], Heiko Runz[7], Sally John[7], Robert Plenge[8], Joseph Maranville[8], George Okafo[9], Nathan Lawless[9], Heli Salminen-Mankonen[9], Mark McCarthy[10], Julie Hunkapiller[10], Meg Ehm[11], Kirsi Auro[11], Simonne Longerich[12], Caroline Fox[12], Anders Mälarstig[13], Katherine Klinger[14], Deepak Raipal[14], Eric Green[15], Robert Graham[15], Robert Yang[16] & Chris O'Donnell[17]

**University of Helsinki & Biobanks** Tomi Mäkelä[18], Jaakko Kaprio[1], Petri Virolainen[19,20,21], Antti Hakanen[19,20,21], Terhi Kilpi[22], Markus Perola[22], Jukka Partanen[23], Anne Pitkäranta[24], Juhani Junttila[25,26,27], Raisa Serpi[25,26,27], Tarja Laitinen[28,29,30], Veli-Matti Kosma[31,32,33], Arto Mannermaa[31,32,33], Jari Laukkanen[34,35,36] & Marco Hautalahti[37]

**Other Experts/Non-Voting Members** Outi Tuovila[38] & Raimo Pakkanen[38]

**Scientific Committee**

**Pharmaceutical companies** Bridget Riley-Gills[5], Jeffrey Waring[5], Ioanna Tachmazidou[6], Chia-Yen Chen[7], Heiko Runz[7], Shameek Biswas[8], Zhihao Ding[9], Marc Jung[9], Rion Pendergrass[10], Julie Hunkapiller[10], Meg Ehm[11], David Pulford[11], Neha Raghavan[12], Adriana Huertas-Vazquez[12], Jae-Hoon Sul[12], Anders Mälarstig[13], Xinli Hu[13], Katherine Klinger[14], Eric Green[15], Robert Graham[15], Sahar Mozaffari[15], Dawn Waterworth[16], Nicole Renaud[17] & Ma´en Obeidat[17]

**University of Helsinki & Biobanks** Samuli Ripatti[1], Johanna Schleutker[19,20,21], Markus Perola[22], Mikko Arvas[23], Olli Carpén[24], Reetta Hinttala[25,26,27], Johannes Kettunen[25,26,27], Katriina Aalto-Setälä[28,29,30], Mika Kähönen[28,29,30], Arto Mannermaa[31,32,33], Jari Laukkanen[34,35,36] & Johanna Mäkelä[37]

**Clinical Groups**

**Neurology Group** Reetta Kälviäinen[33], Valtteri Julkunen[33], Hilkka Soininen[33], Anne Remes[27], Mikko Hiltunen[33], Jukka Peltola[30], Pentti Tienari[39], Juha Rinne[21], Roosa Kallionpää[21], Ali Abbasi[5], Adam Ziemann[5], Jeffrey Waring[5], Sahar Esmaeeli[5], Nizar Smaoui[5], Anne Lehtonen[5], Susan Eaton[7], Heiko Runz[7], Sanni Lahdenperä[7], Janet van Adelsberg[8], Shameek Biswas[8], Julie Hunkapiller[10], Natalie Bowers[10], Edmond Teng[10], Sarah Pendergrass[10], Onuralp Soylemez[12], Kari Linden[13], Fanli Xu[11], David Pulford[11], Kirsi Auro[11], Laura Addis[11], John Eicher[11], Minna Raivio[39], Sarah Pendergrass[10], Beryl Cummings[15] & Juulia Partanen[1]

**Gastroenterology Group** Martti Färkkilä[39], Jukka Koskela[39], Sampsa Pikkarainen[39], Airi Jussila[30], Katri Kaukinen[30], Timo Blomster[27], Mikko Kiviniemi[33], Markku Voutilainen[21], Ali Abbasi[5], Graham Heap[5], Jeffrey Waring[5], Nizar Smaoui[5], Fedik Rahimov[5], Anne Lehtonen[5], Keith Usiskin[8], Tim Lu[10], Natalie Bowers[10], Danny Oh[10], Sarah Pendergrass[10], Kirsi Kalpala[13], Melissa Miller[13], Xinli Hu[13], Linda McCarthy[11], Onuralp Soylemez[12] & Mark Daly[1]

**Rheumatology Group** Kari Eklund[39], Antti Palomäki[21], Pia Isomäki[30], Laura Pirilä[21], Oili Kaipiainen-Seppänen[33], Johanna Huhtakangas[27], Ali Abbasi[5], Jeffrey Waring[5], Fedik Rahimov[5], Apinya Lertratanakul[5], Nizar Smaoui[5], Anne Lehtonen[5], David Close[6], Marla Hochfeld[8], Natalie Bowers[10], Sarah Pendergrass[10], Onuralp Soylemez[12],

Kirsi Kalpala[13], Nan Bing[13], Xinli Hu[13], Jorge Esparza Gordillo[11], Kirsi Auro[11], Dawn Waterworth[16] & Nina Mars[1]

**Pulmonology Group** Tarja Laitinen[30], Margit Pelkonen[33], Paula Kauppi[39], Hannu Kankaanranta[30], Terttu Harju[27], Riitta Lahesmaa[21], Nizar Smaoui[5], Alex Mackay[6], Glenda Lassi[6], Susan Eaton[7], Steven Greenberg[8], Hubert Chen[10], Sarah Pendergrass[10], Natalie Bowers[10], Joanna Betts[11], Soumitra Ghosh[11], Kirsi Auro[11], Rajashree Mishra[11] & Sina Rüeger[1]

**Cardiometabolic Diseases Group** Teemu Niiranen[40], Felix Vaura[40], Veikko Salomaa[40], Markus Juonala[21], Kaj Metsärinne[21], Mika Kähönen[30], Juhani Junttila[27], Markku Laakso[33], Jussi Pihlajamäki[33], Daniel Gordin[39], Juha Sinisalo[39], Marja-Riitta Taskinen[39], Tiinamaija Tuomi[39], Jari Laukkanen[36], Benjamin Challis[6], Dirk Paul[6], Julie Hunkapiller[10], Natalie Bowers[10], Sarah Pendergrass[10], Onuralp Soylemez[12], Jaakko Parkkinen[13], Melissa Miller[13], Russell Miller[13], Audrey Chu[11], Kirsi Auro[11], Keith Usiskin[8], Amanda Elliott[1,4], Joel Rämö[1], Samuli Ripatti[1], Mary Pat Reeve[1] & Sanni Ruotsalainen[1]

**Oncology Group** Tuomo Meretoja[39], Heikki Joensuu[39], Olli Carpén[39], Lauri Aaltonen[39], Johanna Mattson[39], Annika Auranen[30], Peeter Karihtala[27], Saila Kauppila[27], Päivi Auvinen[33], Klaus Elenius[21], Johanna Schleutker[21], Relja Popovic[5], Jeffrey Waring[5], Bridget Riley-Gillis[5], Anne Lehtonen[5], Jennifer Schutzman[10], Julie Hunkapiller[10], Natalie Bowers[10], Sarah Pendergrass[10], Andrey Loboda[12], Aparna Chhibber[12], Heli Lehtonen[13], Stefan McDonough[13], Marika Crohns[14], Sauli Vuoti[14], Diptee Kulkarni[11], Kirsi Auro[11], Esa Pitkänen[1], Nina Mars[1] & Mark Daly[1]

**Opthalmology Group** Kai Kaarniranta[33], Joni A. Turunen[39], Terhi Ollila[39], Sanna Seitsonen[39], Hannu Uusitalo[30], Vesa Aaltonen[21], Hannele Uusitalo-Järvinen[30], Marja Luodonpää[27], Nina Hautala[27], Mengzhen Liu[5], Heiko Runz[7], Stephanie Loomis[7], Erich Strauss[10], Natalie Bowers[10], Hao Chen[10], Sarah Pendergrass[10], Anna Podgornaia[12], Juha Karjalainen[1,4] & Esa Pitkänen[1]

**Dermatology Group** Kaisa Tasanen[27], Laura Huilaja[27], Katariina Hannula-Jouppi[39], Teea Salmi[30], Sirkku Peltonen[21], Leena Koulu[21], Kirsi Kalpala[13], Ying Wu[13], David Choy[10], Sarah Pendergrass[10], Nizar Smaoui[5], Fedik Rahimov[5], Anne Lehtonen[5] & Dawn Waterworth[16]

**Odontology Group** Pirkko Pussinen[39], Aino Salminen[39], Tuula Salo[39], David Rice[39], Pekka Nieminen[39], Ulla Palotie[39], Juha Sinisalo[39], Maria Siponen[33], Liisa Suominen[33], Päivi Mäntylä[33], Ulvi Gursoy[21], Vuokko Anttonen[27], Kirsi Sipilä[27] & Sarah Pendergrass[10]

**Women's Health and Reproduction Group** Hannele Laivuori[1], Venla Kurra[30], Oskari Heikinheimo[39], Ilkka Kalliala[39], Laura Kotaniemi-Talonen[30], Kari Nieminen[30], Päivi Polo[21], Kaarin Mäkikallio[21], Eeva Ekholm[21], Marja Vääräsmäki[27], Outi Uimari[27], Laure Morin-Papunen[27], Marjo Tuppurainen[33], Katja Kivinen[1], Elisabeth Widén[1], Taru Tukiainen[1], Mary Pat Reeve[1], Mark Daly[1], Liu Aoxing[1], Eija Laakkonen[35], Niko Välimäki[41], Lauri Aaltonen[39], Johannes Kettunen[27], Mikko Arvas[42], Jeffrey Waring[5], Bridget Riley-Gillis[5], Mengzhen Liu[5], Janet Kumar[11], Kirsi Auro[11], Andrea Ganna[1] & Sarah Pendergrass[10]

**FinnGen Analysis working group** Justin Wade Davis[5], Bridget Riley-Gillis[5], Danjuma Quarless[5], Fedik Rahimov[5], Sahar Esmaeeli[5], Slavé Petrovski[6], Eleonor Wigmore[6], Adele Mitchell[7], Benjamin Sun[7], Ellen Tsai[7], Denis Baird[7], Paola Bronson[7], Ruoyu Tian[7], Stephanie Loomis[7], Yunfeng Huang[7], Joseph Maranville[8], Shameek Biswas[8], Elmutaz Mohammed[8], Samir Wadhawan[8], Erika Kvikstad[8], Minal Caliskan[8], Diana Chang[10], Julie Hunkapiller[10],

Tushar Bhangale[10], Natalie Bowers[10], Sarah Pendergrass[10], Kirill Shkura[12], Victor Neduva[12], Xing Chen[13], Åsa Hedman[13], Karen S. King[11], Padhraig Gormley[11], Jimmy Liu[11], Clarence Wang[14], Ethan Xu[14], Franck Auge[14], Clement Chatelain[14], Deepak Rajpal[14], Dongyu Liu[14], Katherine Call[14], Tai-He Xia[14], Beryl Cummings[15], Matt Brauer[15], Huilei Xu[17], Amy Cole[17], Jonathan Chung[17], Jaison Jacob[17], Katrina de Lange[17], Jonas Zierer[17], Mitja Kurki[1,4], Samuli Ripatti[1], Mark Daly[1], Juha Karjalainen[1,4], Aki Havulinna[1], Juha Mehtonen[1], Priit Palta[1], Shabbeer Hassan[1], Pietro Della Briotta Parolo[1], Wei Zhou[4], Mutaamba Maasha[4], Shabbeer Hassan[1], Susanna Lemmelä[1], Manuel Rivas[43], Aarno Palotie[1], Arto Lehisto[1], Andrea Ganna[1], Vincent Llorens[1], Hannele Laivuori[1], Mari E. Niemi[1], Taru Tukiainen[1], Mary Pat Reeve[1], Henrike Heyne[1], Nina Mars[1], Kimmo Palin[41], Javier Garcia-Tabuenca[29], Harri Siirtola[29], Tuomo Kiiskinen[1], Jiwoo Lee[1,4], Kristin Tsuo[1,4], Amanda Elliott[1,4], Kati Kristiansson[22], Mikko Arvas[23], Kati Hyvärinen[42], Jarmo Ritari[42], Miika Koskinen[24], Olli Carpén[24], Johannes Kettunen[25,26,27], Katri Pylkäs[26], Marita Kalaoja[26], Minna Karjalainen[26], Tuomo Mantere[25,26,27], Eeva Kangasniemi[28,29,30], Sami Heikkinen[32], Arto Mannermaa[31,32,33], Eija Laakkonen[35], Samuel Heron[20], Dhanaprakash Jambulingam[20], Venkat Subramaniam Rathinakannan[20] & Nina Pitkänen[19,20,21]

**Biobank directors** Perttu Terho[19,20,21], Sirpa Soini[22], Jukka Partanen[23], Eero Punkka[24], Raisa Serpi[25,26,27], Sanna Siltanen[28,29,30], Veli-Matti Kosma[31,32,33] & Teijo Kuopio[34,35,36]

**FinnGen Teams**

**Administration** Anu Jalanko[1], Huei-Yi Shen[1], Risto Kajanne[1] & Mervi Aavikko[1]

**Analysis** Mitja Kurki[1,4], Juha Karjalainen[1,4], Pietro Della Briotta Parolo[1], Arto Lehisto[1], Juha Mehtonen[1], Wei Zhou[4], Masahiro Kanai[4] & Mutaamba Maasha[4]

**Clinical Endpoint Development** Hannele Laivuori[1], Aki Havulinna[1], Susanna Lemmelä[1], Tuomo Kiiskinen[1] & L. Elisa Lahtela[1]

**Communication** Mari Kaunisto[1]

**E-Science** Elina Kilpeläinen[1], Timo P. Sipilä[1], Oluwaseun Alexander Dada[1], Awaisa Ghazal[1] & Anastasia Kytölä[1]

**Genotyping** Kati Donner[1] & Timo P. Sipilä[1]

**Sample Collection Coordination** Anu Loukola[24]

**Sample Logistics** Päivi Laiho[22], Tuuli Sistonen[22], Essi Kaiharju[22], Markku Laukkanen[22], Elina Järvensivu[22], Sini Lähteenmäki[22], Lotta Männikkö[22] & Regis Wong[22]

**Registry Data Operations** Minna Brunfeldt[22], Kati Kristiansson[22], Susanna Lemmelä[1], Sami Koskelainen[22], Tero Hiekkalinna[22] & Teemu Paajanen[22]

**Sequencing Informatics** Priit Palta[1], Kalle Pärn[1], Shuang Luo[1] & Vishal Sinha[1]

**Trajectory** Tarja Laitinen[30], Mary Pat Reeve[1], Harri Siirtola[29], Javier Gracia-Tabuenca[29], Mika Helminen[29], Tiina Luukkaala[29] & Iida Vähätalo[29]

**Data protection officer** Jyrki Pitkänen[1]

**FinBB - Finnish biobank cooperative** Marco Hautalahti[37], Mirkka Koivusalo[37], Sarah Smith[37] &
Tom Southerington[37]

[5]Abbvie, Chicago, IL, USA. [6]Astra Zeneca, Cambridge, UK. [7]Biogen, Cambridge, MA, USA. [8]Bristol Myers Squibb, New York, NY, USA. [9]Boehringer Ingelheim International GmbH, Ingelheim, Germany. [10]Genentech, San Francisco, CA, USA. [11]GlaxoSmithKline, Brentford, UK. [12]Merck, Kenilworth, NJ, USA. [13]Pfizer, New York, NY, USA. [14]Sanofi, Paris, France. [15]Maze Therapeutics, San Francisco, CA, USA. [16]Janssen Biotech, Beerse, Belgium. [17]Novartis, Basel, Switzerland. [18]HiLIFE, University of Helsinki, Helsinki, Finland. [19]Auria Biobank, University of Turku, Hospital District of Southwest Finland, Turku, Finland. [20]University of Turku, Turku, Finland. [21]Hospital District of Southwest Finland, Turku, Finland. [22]THL Biobank/The Finnish Institute for Health and Welfare, Helsinki, Finland. [23]Finnish Red Cross Blood Service, Finnish Hematology Registry and Clinical Biobank, Helsinki, Finland. [24]Helsinki Biobank/Helsinki University and Hospital District of Helsinki and Uusimaa, Helsinki, Finland. [25]Northern Finland Biobank Borealis, University of Oulu, Northern Ostrobothnia Hospital District, Oulu, Finland. [26]University of Oulu, Oulu, Finland. [27]Northern Ostrobothnia Hospital District, Oulu, Finland. [28]Finnish Clinical Biobank Tampere, University of Tampere, Pirkanmaa Hospital District, Tampere, Finland. [29]University of Tampere, Tampere, Finland. [30]Pirkanmaa Hospital District, Tampere, Finland. [31]Biobank of Eastern Finland, University of Eastern Finland, Northern Savo Hospital District, Kuopio, Finland. [32]University of Eastern Finland, Kuopio, Finland. [33]Northern Savo Hospital District, Kuopio, Finland. [34]Central Finland Biobank, University of Jyväskylä, Central Finland Health Care District, Jyväskylä, Finland. [35]University of Jyväskylä, Jyväskylä, Finland. [36]Central Finland Health Care District, Jyväskylä, Finland. [37]Finnish Biobank Cooperative – FINBB, Helsinki, Finland. [38]Business Finland, Helsinki, Finland. [39]Hospital District of Helsinki and Uusimaa, Helsinki, Finland. [40]The National Institute of Health and Welfare, Helsinki, Finland. [41]University of Helsinki, Helsinki, Finland. [42]Finnish Red Cross Blood Service, Helsinki, Finland. [43]University of Stanford, Stanford, CA, USA.

