## [Transparent Peer Review File · Communications Biology]

Peer Review Information

Reviewer comments, first version:

Reviewer #1 (Remarks to the Author: Overall significance):

The authors present straightforward and rigorously derived and presented polygenically-informed cardiometabolic disease prediction models. The models perform as-well to somewhat better than previously described scores with some interesting gains in net reclassification and detection of incident events. The authors state they are able to make these predictions via genetics + survey without the measure of clinical risk factors - but some of the survey questions are dependent upon the measurement of those risk factors (i.e. being on a statin or blood pressure medication). Regardless, as a research tool, which seems to be the intent of the authors, there are some advantages to the simplicity of the query. I would be interested to know if the GRIT-T2D score could contribute to CHD prediction. Overall, a well presented derivation of a novel risk score with some benefits in research implementation.

Reviewer #1 (Remarks to the Author: Impact):

Nature Communications seems fine. There is nothing terribly novel here - the authors applied a PRS derivation tool to their cohort and validated in another (with inclusion of clinical risk). Nothing novel in terms of predictors and thus no great resultant gain in power. There are some nice details for model derivation - that could be a nice reference for the field.

Reviewer #1 (Remarks to the Author: Strength of the claims):

The claims are fine but the authors are not claiming anything dramatic. The claim of the questions being survey based only are off-base as a clinical tool as you would need to have measured LDL or bp to have medication prescribed. But as a data gathering research tool the approach makes sense. I feel this is the likely intention, to help with data mining across cohorts.

Reviewer #1 (Remarks to the Author: Reproducibility):

Independent replication is the gold standard. These are some very European cohorts - but they've made no claim to generalizability.

Reviewer #2 (Remarks to the Author: Overall significance):

Many thanks for the opportunity to review this manuscript.

Tamlander et al. have developed new genomics-enhanced risk tools (GRIT) for CHD and T2D by integrating genetic risk scores with simple risk factors from online questionnaires, without additional laboratory tests.

The authors have derived the prediction models in the FinnGen study, and have independently validated them in the UK Biobank. The models have shown comparable predictive performance to some well-known clinical risk scores (QRISK 3 and QDiabetes).

Reviewer #2 (Remarks to the Author: Strength of the claims):

1. Line 21 - 30, the baseline models are PRS + age / PRS+age+BMI. Since the whole point of the paper is PRS + simple risk factors, I would also recommend showing the prediction of PRS alone and simple risk factors alone.

2. Figure 2, the authors have shown the performance of QRISK3 is worse than GRIT-CHD+ or even GRIT-CHD. From Supplementary Table 4, it seems all variables in QRISK3 except PRS were included in GRIT-CHD+ / GRIT-CHD. Is the higher prediction value of GRIT-CHD and GRIT-CHD+ mainly due to PRS? This should be discussed.

Reviewer #2 (Remarks to the Author: Reproducibility):

1. In supplementary Table 2A 2B, and 3A 3B, the authors have shown the participant characteristics for excluded individuals, eg. in STable 3B, a relatively large proportion of participants was excluded due to the missing HDL-C levels ($43829/121113 = 36\%$). It might be worth exploring the characteristics of participants with missing values. Would imputation provide a more reasonable validation dataset compared with excluding missing values?

2. The authors mentioned that the input weight samples partly overlap with FinnGen study and are independent of UKB. The effect on training the PRS model should be discussed.

3. The validation UKB data for CHD and T2D are from different sources, eg. ICD10/9 hospital inpatient records, OPCS4, medication, general practice codes, and self-reported codes in 20002. They have different coverage of UKB participants, eg, general practice Read V2 and Read V3 clinical codes cover half UKB samples. Will this affect the validation step for GRIT - CHD and T2D.

Reviewer #3 (Remarks to the Author: Overall significance):

This manuscript describes the combination of new PRS and clinical assessments for prediction of incident CAD and T2D. The paper is clearly written and comparatively exhaustive in explanation, though some holes remain in my view, as mentioned below. Positives include the use of PRS-CS to generate best-in-class PRS, out-of-sample validation in the complete British UKBB, and demonstration that blood pressure, cholesterol, and triglycerides as continuous traits improve joint modeling.

However, I have to say that I am struggling to see what is new here. Multiple studies cited in the manuscript (including one from the second author) and others have established both that PRS are strongly predictive of incident and prevalent CAD or T2D, and that at least for CAD the PRS add surprisingly little to the established clinical factors after middle age. The title of the manuscript implies that simple questionnaires are adequate to capture these clinical factors, which may be novel, but the strongest incremental improvement is with BP, HDL and TG, which are not questionnaire-based measures.

Reviewer #3 (Remarks to the Author: Strength of the claims):

In my opinion the authors overstate the case for improvement due to PRS to a high degree. Although the GRIT scores are significantly higher than Baseline PRS+age+sex as well as PCE/QRISK3/QDiabetes in Figure 2, Fig S3 places this in perspective: the increment is in no way clinically meaningful as the ROC almost overlay. The OR are also significantly higher, but the relevant clinical measure is the NRI, which is also improved by several percent BUT at the cost of a large number of reverses classifications. Improving the classification for 20 people while decrementing it for 15 results in a net improvement but at substantial cost. Table 2 presents the numbers, but it is not transparent, and shows clearly that 10x to 20X non-cases are reclassified than cases.

I think the message of the paper would be much clearer with a statement not just of NRI but of the number reclassified to achieve this net gain – something analogous to the number needed to treat.

A focus on the performance of PRS solely in the high risk (young, before onset of the clinical factors; female, normoweight) should provide a stronger statement of utility. The data presented seem to reaffirm that PRS have little added utility in moderate or high clinical risk individuals.

Minor Points:

1. Numbers for Table 1 Derivation CHD do not add up: $30,836+28,104 \neq 61,878$

2. Does age refer to age at enrolment or incidence or endpoint?
3. In the Statistical Analysis section (page 9 line 12) maximum follow-up time is stated as 10 years from baseline (enrolment?), but on the next page in the Results the median follow-up time was 15.3 years. Please explain.
4. I understand the rationale for removing prevalent cases at Baseline, but can the authors estimate the impact on under-estimation of prevalence and PRS performance in a naïve (ideally, younger) population. The genetics should be stronger in the removed prevalent cases, but disease is also less prevalent.
5. Please provide more details on the calibration: do not assume that readers understand the concepts of baseline hazard and mean component, or how they were used for recalibration. This is really important in my opinion since calibration is critical but poorly discussed or understood in the literature.
6. Accordingly, I am concerned that the prevalence in FinnGen (Table 1) is ~3X greater for both traits in Finland than the UK. Quite apart from this being surprising to me, it must impact PRS assessment, but intuitively (correct me if I am wrong) in the opposite direction to that shown in Fig 2. Higher prevalence generally implies lower OR per Falconer, so applying the Finn OR to UKBB ought to underestimate prevalence, but apparently predicted risk is greatly elevated in the UKBB. Also, there does not appear to have been an adjustment for Finnish ancestry. These issues should be discussed for clarity.

Reviewer #3 (Remarks to the Author: Reproducibility):

Reproducibility of methods is fine, but as I understand it few researchers have open access to Finnish databanks - though it can be arranged as a collaboration.

I do not feel that all of the Supplementary Data is necessary, so in that sense they have gone overboard in promoting repeatability.

Depositing the PRS-CS scores in the public database on publication is really good.

Author rebuttal, first version:

Please find below our point-by-point responses to the Reviewers' comments. We thank the Reviewers for the constructive and insightful feedback, which has helped to improve the manuscript considerably. The Reviewers raised several important points which helped to clarify and improve the quality and reproducibility of the study. The main changes include updated analyses accounting for Finnish ancestry through adjusting with 10 first principal components of ancestry, improvements in reporting of predictive performance across individual risk factors included in the GRIT scores, and evaluating how different data sources may impact the validation of the GRIT scores. We have also expanded the discussion with respect to limitations, generalizability, and clinical utility of the GRIT scores.

Reviewers' Comments:

Reviewer #1: The authors present straightforward and rigorously derived and presented polygenically-informed cardiometabolic disease prediction models. The models perform as well to somewhat better than previously described scores with some interesting gains in net reclassification and detection of incident events. The authors state they are able to make these predictions via genetics + survey without the measure of clinical risk factors - but some of the survey questions are dependent upon the measurement of those risk factors (i.e. being on a statin or blood pressure medication). Regardless, as a research tool, which seems to be the intent of the authors, there are some advantages to the simplicity of the query. Overall, a wellpresented derivation of a novel risk score with some benefits in research implementation.

Author response: We would like to thank Reviewer #1 for the encouraging feedback. Our GRIT-CHD and GRIT-T2D scores show at least similar discrimination and notable gains in net reclassification compared to standard clinical risk scores for CHD and T2D. The simplicity of the approach allows for practical and relatively effortless implementation of the GRIT scores across resources where genome data is available. We agree that although some of the survey questions indirectly rely on previous laboratory measurements, they are still generally obtainable by simple surveys or from electronic health records. We believe that our scores could be used as either a research tool, or if PRSs were widely available in future health systems, such combination scores could serve in preselecting individuals for more frequent or targeted health checkups. In these checkups, the risk could then be further assessed through laboratory measurements and established clinical risk calculators.

Specific comments:

1: I would be interested to know if the GRIT-T2D score could contribute to CHD prediction.

Author response: The risk of CHD and T2D is often assessed separately, which is why the GRIT-CHD and GRIT-T2D scores use disease-specific predictors. However, as many of the predictors (e.g., BMI, smoking status, age) are included in both scores, we agree that this is an interesting aspect to test. The Pearson correlation between the GRIT-CHD and GRIT-T2D was 0.42 in incident CHD analyses and 0.47 in incident T2D analyses. Similarly, the Pearson correlation between GRIT-CHD+ and GRIT-T2D+ was 0.54 in incident CHD analyses

and 0.55 in incident T2D analyses. The Pearson correlations were statistically significant ($P < 2.2 \times 10^{-16}$ for all comparisons). We have now added a new Supplementary Figure 7 (below for reference) which shows how GRIT-CHD and GRIT-T2D work in cross-prediction.

Supplementary Figure 7. Cross-predictive performance of the Genomics-enhanced Risk Tools (GRIT) for CHD and T2D. GRIT-CHD is tested in prediction of incident T2D, and GRIT-T2D in prediction of incident CHD. The sample size in CHD is slightly smaller ($N = 242,565$ with 4,467 incident cases) than in the main analyses due to exclusions of individuals with missing values for TG needed for GRIT-T2D+. Similarly, in T2D, the sample size becomes slightly smaller due to exclusions of missing values for LDL, resulting in a sample size of 120,951 with 2,542 incident cases.

Following from the positive correlation of the two scores (and the importance of the shared key common risk factors in both diseases), the cross-disease predictive performance in both diseases was not insignificant, but though it was much lower than with the disease-specific predictions for CHD and T2D. For incident CHD, the GRIT-CHD had a higher AUC (AUC = 0.774; 95% CI: 0.768 to 0.780) compared to both the GRIT-T2D and GRIT-T2D+. For incident T2D, the GRIT-T2D had a higher AUC (AUC = 0.811; 95% CI: 0.803 to 0.818) compared to both the GRIT-CHD and GRIT-CHD+.

We have now modified the Results on page 12 to read as follows:

“The cross-predictive performance of the GRIT scores is shown in Supplementary Figure 7”.

2: *Nature Communications* seems fine. There is nothing terribly novel here - the authors applied a PRS derivation tool to their cohort and validated in another (with inclusion of clinical risk). Nothing novel in terms of predictors and thus no great resultant gain in power. There are some nice details for model derivation - that could be a nice reference for the field.

Author response: We have answered comments 2 and 3 jointly.

3: The claims are fine but the authors are not claiming anything dramatic. The claim of the questions being survey based only are off-base as a clinical tool as you would need to have

measured LDL or bp to have medication prescribed. But as a data gathering research tool the approach makes sense. I feel this is the likely intention, to help with data mining across cohorts.

Author response: We have answered comments 2 and 3 jointly. CHD and T2D are highly prevalent diseases with well-established risk factors, clinical risk calculators, and preventative strategies in clinical practice (PMIDs 24222018, 27222591, 33298414). Particularly CHD has been widely studied for the impact of PRSs on disease risk. With this premise, finding novel predictors and large gains in predictive power is a challenge. However, as our goal was to develop simple tools that are easy to use particularly in situations where the genome-wide data is available and can therefore be efficiently used for preselecting individuals to further risk assessment, we believe that non-inferiority to widely used clinical risk calculators is a good achievement.

To our knowledge, such a setting has not been applied previously at large scale. Furthermore, one additional new strength of the study is the use of data obtained from UK general practices – particularly T2D is usually diagnosed and treated in primary care. We agree that although some of the survey questions indirectly rely on previous laboratory measurements, they are generally obtainable by simple surveys or from electronic health records, and do not require immediate laboratory tests.

Based on this comment, we have now revised parts of the Discussion to describe and justify our approach more clearly. The changes in the Discussion on page 14 read as follows:

“In contrast to previous studies, we studied the role of PRSs combined with clinical risk factors obtainable by simple questionnaires, without the need of additional laboratory and clinical measurements such as blood pressure and lipids. With the decreasing costs of sequencing, rapidly expanding knowledge of genomics, and increasing public interest, genome data is becoming increasingly available for applications of disease prevention and care^{1,2}. Our approach allows for efficient and relatively effortless risk estimation across resources where genome data is readily available, but measurements of quantitative risk factors used in clinical care such as lipids might not be. Examples include population-based biobanks and commercial genomics databases, or when clinical variables from healthy, asymptomatic individuals have not yet been measured.”

4: Independent replication is the gold standard. These are some very European cohorts - but they've made no claim to generalizability.

Author response: We agree on the importance of independent, out-of-sample validation and replication in genetic risk prediction studies. One main limitation often overlooked in previous studies has been the lack of multiple datasets for studying external validity, with the risk models often derived and validated in the same cohort sample by dividing the discovery sample into separate derivation and validation samples.

This study uses two large datasets for a cross-biobank approach in evaluating the PRS-enhanced risk tools, covering two different populations and healthcare settings.

Similarly, many validation studies have observed limited external performance when assessing similar risk scores, in contrast to the present study. We believe that the study provides evidence for generalizability to other European cohorts, as the GRIT-CHD and GRIT-T2D scores had good external validity in UK Biobank. Despite a very low response rate in UK Biobank, risk factor associations in the UK Biobank seem to be generalizable (e.g., PMID 32051121).

On page 15 we have stated:

“As the predictive performance of risk models can greatly worsen when assessed outside of their derivation datasets³⁷, our results indicate that GRIT-CHD and GRIT-T2D may generalize well also to other cohorts.”

This study did not assess generalizability to non-European individuals and other healthcare settings, and research is needed on more diverse and larger population settings, which is why we have shared the relevant details on the GRIT-T2D and GRIT-CHD scores for additional external validation studies. We have now modified the limitations on page 15 to read as follows:

“Additionally, our data was limited to middle-aged individuals of European ancestry. External validation of the GRIT-CHD and GRIT-T2D scores is needed for diverse ancestries, including integration of ancestry-specific PRSs. Despite a very low response rate resulting in oversampling of healthy individuals in UK Biobank, the risk factor associations have been considered to be generalizable³⁹.”

5: Pretty basic stuff but well presented with interesting utility. Certainly doesn't rise to the level of *Nature Genetics*. But it's a decent reference point for the field.

Author response: We would like to thank the reviewer. We believe that the approach also has potential to serve as a reference for future studies in other common complex diseases as well.

Reviewer #2: Tamlander et al. have developed new genomics-enhanced risk tools (GRIT) for CHD and T2D by integrating genetic risk scores with simple risk factors from online questionnaires, without additional laboratory tests. The authors have derived the prediction models in the FinnGen study, and have independently validated them in the UK Biobank. The models have shown comparable predictive performance to some well-known clinical risk scores (QRISK 3 and QDiabetes).

Specific comments:

1: Lines 21-30: The baseline models are PRS + age / PRS+age+BMI. Since the whole point of the paper is PRS + simple risk factors, I would also recommend showing the prediction of PRS alone and simple risk factors alone.

Author response: First, we want to thank reviewer #2 for the comments which have considerably improved the manuscript. We presented the prediction of the risk factors jointly, as we felt that would show the general performance of the GRIT scores. We additionally showed the prediction of PRS alone in Supplementary Table 9. However, we agree it is also important to demonstrate in more detail the effects of PRS and additional risk factors.

Based on this comment, we have now modified the main Figure 2 and added a new Figure 3 (both shown below for reference), to 1) assess the individual predictors included in the GRIT scores when combined with age and sex, 2) to assess the impact of GRIT scores without PRSs, and 3) to separately demonstrate the performance of the GRIT scores against clinical risk scores in UK Biobank. Prediction by sex and age subgroups is detailed in Supplementary Figure 1 (Reviewer #3 comment 7).

Figure 2. Area under the receiver operating characteristic curve (AUC) with 95% confidence intervals (CI) for individual risk factors and the Genomics-enhanced Risk Tools (GRIT) in the validation cohort, UK Biobank. Panel (a) shows results for CHD (N = 242,687) and panel (b) for T2D (N = 121,113). The AUC was first calculated for age and sex and additionally for each individual risk factor integrated with age and sex. Lastly, the AUC was calculated for the GRIT scores and the GRIT scores without PRSs. Points indicate AUC estimates and error bars represent the 95% CIs for each risk factor, with incident disease as endpoint. BMI = body mass index; LDL = low-density lipoprotein; HDL = high-density lipoprotein, TG = triglycerides, CVD = cardiovascular disease.

Figure 3. Area under the receiver operating characteristic curve (AUC) with 95% confidence intervals (CI) for the Genomics-enhanced Risk Tools (GRIT) and clinical risk scores in the validation cohort, UK Biobank. Panel (a) shows results for CHD (N = 242,687) and panel (b) for T2D (N = 121,113). The sex-specific baseline models include age and PRS, and additionally BMI for the T2D model. Our sex-specific Genomics-enhanced Risk Tools (GRIT-CHD and GRIT-T2D) were compared to established clinical risk scores (Pooled Cohort Equations and QRISK3 for CHD, FINDRISC and QDiabetes for T2D). Points indicate AUC estimates and error bars represent the 95% CIs for each factor, with incident disease as endpoint. All tests were two-sided. PCE = Pooled Cohort Equations.

In CHD (Figure 2, panel a), the PRS_{CHD} combined with age and sex had a higher AUC (AUC = 0.757, 95% CI 0.750–0.764) for incident CHD than any of the 8 other risk factors (smoking, BMI, use of antihypertensives, diabetes, family history of CHD, systolic blood pressure, LDL and HDL).

In T2D (Figure 2, panel b), the PRS_{T2D} combined with age and sex had a similar or a higher AUC (AUC = 0.673, 95% CI 0.663–0.683) than 7 of the 10 conventional risk factors (smoking, use of statins, CVD, gestational diabetes, family history of T2D, systolic blood pressure, HDL).

Based on this comment, we have now edited the Methods on page 7 to read as follows:

“We separately evaluated the incremental value of each individual risk factor when added on top of with age and sex.”

And the Results on page 10 to read as follows:

“We tested the incremental value of each risk factor by adding them individually to the model with age and sex in the final FinnGen derivation datasets. The AUC increments ranged from +0.002 to +0.03 in CHD and from +0.006 to +0.17 in T2D in UK Biobank (Figure 2, results by sex and age subgroups in Supplementary Figure 1). The largest increments in AUC came from PRS_{CHD} (in CHD analyses) and BMI (in T2D analyses).”

“Next, we compared our new risk scores to routinely used clinical risk calculators: GRIT-CHD to the QRISK3 and Pooled Cohort Equations (PCE) algorithms and the GRIT-T2D to the QDiabetes and FINDRISC algorithms (Figure 3, Supplementary Figure 2). We used sex-

specific baseline models that included age and PRS, and additionally BMI for the T2D model analyses, as benchmarks for the performance of both the clinical risk scores and the GRIT scores. Measured with AUC, the discrimination of the baseline model for CHD was 0.756 (95% CI 0.750–0.763) and for the baseline model for T2D, 0.789 (0.780–0.797).”

and the Discussion on page 13 to read as follows:

”When added on top of with age and sex, the PRS_{CHD} had a higher AUC for incident CHD than any of the eight other individual risk factors of GRIT-CHD. A largely corresponding effect of the PRS_{CHD} was recently demonstrated also among symptomatic patients with suspected CHD³². Similarly, when added on top of with age and sex, the PRS_{T2D} had a similar or a higher AUC for incident T2D than seven of the ten included conventional risk factors. Our GRIT-CHD and GRIT-T2D models showed at least comparable predictive performance with widely applied clinical risk scores for CHD and T2D, and the overall reclassification improvements with GRIT-CHD and GRIT-T2D were driven by improvements particularly in incident disease cases.”

2: Figure 2: the authors have shown the performance of QRISK3 is worse than GRIT-CHD+ or even GRIT-CHD. From Supplementary Table 4, it seems all variables in QRISK3 except PRS were included in GRIT-CHD+ / GRIT-CHD. Is the higher prediction value of GRIT-CHD and GRIT-CHD+ mainly due to PRS? This should be discussed.

Author response: The higher prediction value likely arises from multiple differences between QRISK3 and the GRIT scores.

First, the GRIT scores contain PRS_{CHD}. In CHD (Figure 2, panel a) we demonstrate that the PRS_{CHD} has a substantially larger effect size than any of the other included risk factors in the GRIT scores in UK Biobank when integrated with age and sex.

Second, while the difference between QRISK3 and GRIT-CHD was statistically significant (Figure 3, panel a; P = 0.0013), the magnitude of the difference was quite modest: The AUC for QRISK3 was 0.767 and for GRIT-CHD 0.774. The difference in AUC between QRISK3 and GRIT-CHD+ was more notable.

Third, the GRIT-CHD and GRIT-CHD+ scores contain only a subset of variables included in QRISK3. The GRIT-CHD, GRIT-CHD+, and QRISK3 scores contain 8, 11, and 21 predictor variables, respectively. The additional variables in QRISK3 are total cholesterol, HDL cholesterol and SBP (these variables are also included in GRIT-CHD+), SBP variability, ethnicity, Townsend index, history of rheumatoid arthritis, history of atrial fibrillation, history of chronic kidney disease, history of migraine, current use of corticosteroid medication, history of systemic lupus erythematosus, current use of atypical antipsychotic medication, history of severe mental illness, and history of or treatment for erectile dysfunction. The algorithm for QRISK3 also includes fractional polynomial terms for age and body mass index, and additionally interactions with age for BMI, SBP, Townsend index, family history of CHD, treated hypertension, atrial fibrillation, type 1 diabetes, type 2 diabetes, chronic kidney disease, and smoking status. It is possible that the large number of comorbidities, and polynomial and interaction terms in QRISK3 used

to model the complexity of cardiovascular disease (non-linear risk relations, significant interactions) might have resulted in decreased performance in external data compared to a less complex model, such as GRIT-CHD.

Fourth, differences in participant characteristics (such as age and BMI) between the derivation datasets (FinnGen for the GRIT scores, the QResearch database for QRISK3) might partly explain the difference in predictive performance in UK Biobank. For example, the participants in the derivation sample of QRISK3 were on average younger (mean age at enrollment (SD) for women 43.3 years (15.3) and for men 42.6 years (14.0)) compared to the individuals in the derivation sample in FinnGen (mean (SD) ages 51.1 years (10.8) for women and 53.4 years (10.7) for men). The mean ages were 56.4 years (7.9) for women and 56.3 years (8.1) for men in the UK Biobank validation sample. The mean BMI (SD) in the QRISK3 derivation sample was slightly smaller (25.4 (5.1) for women and 25.9 (4.2) for men) than in the FinnGen and UK Biobank samples. The mean (SD) BMI in the FinnGen derivation sample was 26.7 (5.2) for women and 27.0 (4.1) for men and in the UK Biobank validation sample 26.7 (4.9) for women and 27.4 (4.0) for men. These differences in age and BMI distribution might have decreased the performance of QRISK3 in UK Biobank, as they should also impact other covariates in QRISK3. However, the interaction terms for age and BMI included in QRISK3 (as detailed above) should at least partly cover these issues.

We have now modified the Discussion on page 14 to read as follows:

“In addition to PRSs, the higher predictive value of the GRIT scores compared to the clinical risk scores in UK Biobank is also likely to be impacted by differences in the model input variables, model complexity, and participant characteristics between the derivation cohorts”.

3: Supplementary Tables 2A-B & 3A-B: The authors have shown the participant characteristics for excluded individuals, e.g. in Supp. Table 3B, a relatively large proportion of participants was excluded due to the missing HDL-C levels (43829/121113 = 36%). It might be worth exploring the characteristics of participants with missing values. Would imputation provide a more reasonable validation dataset compared with excluding missing values?

Author response: The proportion of individuals excluded due to missing HDL-C in the T2D analyses was accidentally shown as 43,829, when in the T2D dataset (N = 160,338) the missing count is 20,713 (12.9%) for HDL-C. This and the other missing counts in the T2D dataset are now corrected in the updated Supplementary Information.

Supplementary Tables 4 and 5 show the characteristics of the excluded participants (based on prevalent CVD/diabetes, use of statins (in CHD dataset), not having data from general practice (in T2D dataset), or missing data on predictors) are detailed. Previous similar analyses of UK Biobank have also shown that the excluded individuals with missing values share similar main characteristics with the included individuals (e.g., PMID 32068818).

To answer this question more specifically, we examined five main characteristics of the individuals (Table below for reference) with missing values in the validation datasets for CHD (N = 343,672 individuals with PRS_{CHD} available) and T2D (N = 160,338 individuals with PRS_{T2D} and GP data available). The main characteristics do not significantly differ from the characteristics of the included individuals (shown in Supplementary Tables 4 & 5). As some aspects related to the question are already shown in Supplementary Tables 4 & 5, and as the amount of supplementary material is already quite extensive (also noted by reviewer #3 comment 15), we chose to not add this as a supplementary table.

	Included individuals in CHD validation set, N = 242,687	Individuals with missing values in CHD validation set, N = 43,391	Included individuals in T2D validation set, N = 121,113	Individuals with missing values in T2D validation set, N = 31,488
Age, mean \pm SD	56.4 \pm 8.0	57.0 \pm 8.0	57.1 \pm 8.0	57.4 \pm 7.9
Male sex, n (%)	105,439 (43.4)	18,151 (41.8)	55,898 (46.2)	12,848 (40.8)
Current smoker, n (%)	23,921 (9.9)	4,145 (9.6)	11,836 (9.8)	3,586 (11.4)
BMI, kg m ⁻² , mean \pm SD	27.0 \pm 4.6	27.2 \pm 4.8	27.2 \pm 4.5	27.7 \pm 5.0
Incident disease*, n (%)	4,469 (1.8)	1,038 (2.4)	2,544 (2.1)	917 (2.9)

Table. *Prevalent disease cases are excluded.

We agree with the reviewer that although multiple imputation could result in a larger dataset for our analyses, it is often laborious when carefully done. Considering the large size of the current validation datasets (N = 242,687 for CHD and N = 121,113 for T2D), and that the individuals with missing data on variables had essentially similar characteristics compared to the included individuals, we chose not to use imputation to cover missing values in the analyses.

4: The authors mentioned that the input weight samples partly overlap with FinnGen study and are independent of UKB. The effect on training the PRS model should be discussed.

Author response: The overlapping samples (healthy controls from the FINRISK 1992, 1997, 2002 and 2007 cohorts) cannot be accurately ascertained using the FinnGen data, as the identifiers for the overlapping samples are not included in the FinnGen data.

Therefore, as a sensitivity analysis, we excluded FINRISK 1992–2007 participants that were not genotyped with the FinnGen ThermoFisher Axiom custom array. This should remove all overlapping samples; however, we stress that this approach also removes a large number of individuals not included in the input weight samples, and considerably decreases the sample size and number of incident cases in the derivation analyses.

In CHD derivation analyses in FinnGen, the sample size would be reduced from 61,878 individuals to 43,318 individuals. The sample size for men would be reduced from 33,774 to 25,307 (a decrease in incident CHD cases from 2,938 to 2,392) and for women, 28,104 to 18,011 (598 to 379). The number of individuals free of CHD included in the input weight samples was 1,208 (PMID 23202125).

Excluding these participants does not have a major impact on the AUC, and only slightly impacts on the OR per SD of the PRS_{CHD} (Table below for reference). However, we note that the possible inflation of the effect size of the PRS_{CHD} would in downstream analyses decrease the predictive performance of the GRIT scores.

PRS _{CHD}	Incident CHD cases	
	AUC (95% CI)	OR per SD (95% CI)
FinnGen, CHD derivation dataset (N = 61,878 with 3,536 cases)	0.819 (0.812–0.825)	1.58 (1.52–1.65)
– “ – with all possible overlapping samples excluded (N = 43,318 with 2,771 cases)	0.810 (0.803–0.817)	1.45 (1.39–1.52)

Table. The estimates are from logistic regression models adjusted for year of birth, sex, the ten first principal components of ancestry, batch, and genotyping array.

Based on this comment, we have now edited the Methods on pages 6–7 to read as follows:

“The input weights came from two large genome-wide association studies (GWAS) independent of UK Biobank, but with some sample overlap in FinnGen for CHD^{12,13}, as we could not accurately exclude the overlapping samples for CHD (1,208 individuals free of CHD in the FINRISK 1992-2007 cohorts included in FinnGen) from the FinnGen dataset.”

And the Discussion on page 15 to read as follows:

“Finally, as a small number of overlapping input weight samples for the PRS_{CHD} within the FinnGen study possibly inflated the effect size of polygenic risk in the joint modeling, the performance of the GRIT-CHD scores in UK Biobank may have been decreased to some degree.”

5: The validation UKB data for CHD and T2D are from different sources, eg. ICD10/9 hospital inpatient records, OPCS4, medication, general practice codes, and self-reported codes in 20002. They have different coverage of UKB participants, eg. general practice Read V2 and Read V3 clinical codes cover half UKB samples. Will this affect the validation step for GRIT – CHD and T2D?

Author response: To our knowledge, this study is one of the first ones to broadly use the individual-level data in UK Biobank (hospital, mortality, and general practice records, and UK Biobank baseline clinical data). Particularly, combining multiple data sources is a common practice in similar studies, e.g., a similar approach to use general practice data alongside hospital and mortality records was used in the development of the QRISK3 and QDiabetes-2018 algorithms (PMIDs 28536104, 29158232), which are used in clinical practice in the UK.

The impact of utilizing the interim release of GP data should, however, be relatively small in this study, as the number of cases identified using only the GP data increases the incident case count only slightly (2.1% in CHD and 14.7% in T2D,

Supplementary Table 17 below for reference). The current interim data release of primary care data to the UK Biobank unfortunately covers only approximately 40% of the UK Biobank participants at the time of revising the manuscript.

We have now added Supplementary Table 17, which details the identified cases in hospital inpatient/mortality registry data, general practice data, and UK Biobank Nurse interview data, and separately the number of cases identified only in the interim release of GP data.

Event	N cases in HES/mortality data (% of total N cases)	N cases in GP data (% of total N cases)	N cases in Nurse interview data (% of total N cases)	Total N cases	N cases in GP data only (% of total N cases)
Prevalent CVD (N = 343,672 before exclusions)	14,312 (58.6%)	6,986 (28.6%)	21,196 (86.8%)	24,415	784 (3.2%)
Prevalent T2D (N = 160,338 before exclusions)	2,958 (38.3%)	4,968 (64.3%)	7,396 (95.8%)	7,722	96 (1.2%)
Incident CHD (N = 242,687)	4,377 (97.9%)	1,014 (22.7%)	–	4,469	92 (2.1%)
Incident T2D (N = 121,113)	2,169 (85.3%)	1,603 (63.0%)	–	2,544	375 (14.7%)

Supplementary Table 17. The number of identified incident and prevalent CVD, CHD, and T2D cases by data source in the validation cohort, UK Biobank. The last column shows the number of events that were only identified using primary care data.

Moreover, several other diseases which we used as comorbidities (risk factors) in the GRIT scores and the clinical risk scores are also often diagnosed or treated in primary care (such as psychiatric diseases, rheumatoid arthritis, erectile dysfunction). Self-reported disease codes (field 20002) have been widely utilized in previous studies utilizing UK Biobank, as they have been recorded by trained health care professionals at UK Biobank.

We have now modified the Results on page 10 to read as follows:

“The validation datasets for the GRIT scores comprised 242,687 UK Biobank individuals with 4,469 incident cases for CHD and 121,113 individuals with 2,544 incident cases for T2D who met our inclusion criteria (Table 1). Supplementary Table 17 shows the number of incident and prevalent cases by type of event and data source in the validation cohort.”

And the Discussion on page 15 to read as follows:

“The impact of utilizing primary care data in this study should be relatively small, as the number of CHD and T2D cases identified using only the general practice data was small. However, the primary care data allowed us to calculate the QRISK3 and QDiabetes clinical scores more accurately as they were derived using the same data sources^{22,23}.”

Reviewer #3: This manuscript describes the combination of new PRS and clinical assessments for prediction of incident CAD and T2D. The paper is clearly written and comparatively exhaustive in explanation, though some holes remain in my view, as mentioned below. Positives include the use of PRS-CS to generate best-in-class PRS, out-of-sample validation in the complete British UKBB, and demonstration that blood pressure, cholesterol, and triglycerides as continuous traits improve joint modeling.

Author response: We appreciate considerably the feedback and suggestions from Reviewer #3, which have provided more rigor to our manuscript.

Specific comments:

Major concerns:

1: However, I have to say that I am struggling to see what is new here. Multiple studies cited in the manuscript (including one from the second author) and others have established both that PRS are strongly predictive of incident and prevalent CAD or T2D, and that at least for CAD the PRS add surprisingly little to the established clinical factors after middle age.

Author response: We would like to also refer to our response to Reviewer #1 comment 3. The GRIT scores could be used to identify people who may be at high risk for CHD and T2D by improving state-of-the-art genome-wide PRSs for CHD and T2D only by a simple set of clinical questions. Our approach resulted in non-inferior, or even better predictive performance compared to conventional clinical scores for CHD and T2D. This is particularly important in CHD, where the QRISK3 score that is derived and validated for use in the UK population had similar performance as the GRIT-CHD, which we derived in a Finnish sample. Therefore, PRSs measuring germline genetic susceptibility for CHD and T2D combined with simple clinical questions has promising utility in preliminary cardiometabolic risk assessment.

Our approach has many practical implications as the GRIT scores do not require deep phenotyping for clinical risk factors. As more genomics resources with larger sample sizes and more diverse ancestry groups become available, the GRIT scores and similar tools can be easily applied across different cohorts, and we have made the weights for the PRSs and the GRIT scores available for this purpose. In addition to our novel approach, we cautiously assess the predictive models, and use updated UK Biobank hospital inpatient, mortality, and newly made available general practice data. We use PRS-CS, a contemporary software for constructing the PRSs. We also show that joint modeling of continuous clinical risk factors (blood pressure and blood lipid measurements) further improves the prediction of the GRIT scores considerably. We have now clarified the Discussion to emphasize our view on the novelty of the study.

We have now modified the Discussion on page 14 to read as follows in addition to the modifications detailed in answers to reviewers #1 and #2:

“In contrast to previous studies, we studied the role of PRSs combined with clinical risk factors obtainable by simple questionnaires, without the need of additional laboratory

and clinical measurements such as blood pressure and lipids. With the decreasing costs of sequencing, rapidly expanding knowledge of genomics, and increasing public interest, genome data is becoming increasingly available for applications of disease prevention and care^{1,2}”

2: The title of the manuscript implies that simple questionnaires are adequate to capture these clinical factors, which may be novel, but the strongest incremental improvement is with BP, HDL and TG, which are not questionnaire-based measures.

Author response: We would consider the manuscript title *“Improving polygenic risk assessment with simple questionnaire-based risk factors in coronary heart disease and type 2 diabetes”* to reflect our primary approach well. As more genomics resources become available across biobanks and populations, simple tools such as our GRIT scores could be used for preselecting individuals for further, detailed risk assessment, which would include measurement of blood pressure and lipids. We therefore highlight that integrating PRSs with simple questionnaire-based risk factors results in non-inferior risk assessment compared to standard clinical risk scores. As blood pressure, LDL, HDL, and TG measurements provided the strong improvements in performance, we also aimed to show the impact of integrating clinical risk factors to the GRIT scores (GRIT-CHD+ and GRIT-T2D+) (Figure 2).

3: In my opinion the authors overstate the case for improvement due to PRS to a high degree. Although the GRIT scores are significantly higher than Baseline PRS+age+sex as well as PCE/QRISK3/QDiabetes in Figure 2, Fig S3 places this in perspective: the increment is in no way clinically meaningful as the ROC almost overlay.

Author response: We have now updated the main Figure 2 (reviewer #2, comment 1) to demonstrate the effect of the PRSs without joint modeling and the GRIT scores without PRSs to demonstrate the effects of PRSs and additional risk factors in more detail. Figure 2 demonstrates that the PRS_{CHD} improves the AUC from 0.72 (Age and sex) to 0.76 (Age and sex + PRS_{CHD}) in CHD resulting in a higher AUC than any of the eight other included conventional risk factors combined with age and sex. Similarly, the PRS_{T2D} improves the AUC from 0.61 (Age and sex) to 0.67 (Age and sex + PRS_{T2D}), and had a similar or a higher AUC for incident T2D than seven of the ten included conventional risk factors when integrated with age and sex.

Our focus for this study was to demonstrate the utility of enhancing PRS-based risk prediction by a simple set of clinical questions, easily obtainable for instance by an online survey, in situations where genome-wide data is readily available, and we think non-inferiority to clinical risk scores in an external cohort is a good achievement.

4: The OR are also significantly higher, but the relevant clinical measure is the NRI, which is also improved by several percent BUT at the cost of a large number of reverses classifications. Improving the classification for 20 people while decrementing it for 15 results in a net improvement but at substantial cost.

Author response: We have answered comments 4 and 5 jointly.

5: Table 2 presents the numbers, but it is not transparent, and shows clearly that 10x to 20x non-cases are reclassified than cases.

Author response: We have answered comments 4 and 5 jointly. While we agree that the GRIT scores overestimate risk for a small proportion of non-cases which leads to somewhat poorer classification, the proportion of individuals non-cases misclassified is still quite small. For instance, in Table 2, comparing the GRIT-CHD to PCE at a risk threshold of 7.5%, the GRIT-CHD upclassifies 5,463 (2.3%) non-cases and downclassifies 3,864 (1.6%) non-cases, resulting in 1,599 (0.7%) non-cases being incorrectly classified to a higher risk class. For incident cases, the GRIT-CHD upclassifies 428 (9.6%) cases and downclassifies 221 (4.9%) cases. The classification movements in T2D are larger.

We think the number of false positives is acceptable, as the GRIT scores were derived as preliminary risk indicators for large cohorts of people with genome data available, for targeting high-risk individuals in need of detailed clinical risk assessment. However, we note that the GRIT scores and the clinical risk scores had mostly similar classifications (Table 2), highlighting that the included predictor variables adequately capture at least some of the effects of BP and lipid measurements not included in the GRIT scores. Furthermore, clinical risk scores used in clinical practice have been shown to misclassify individuals by overestimating absolute risk, particularly in older individuals (e.g., PMID 30954144, 27436865).

In Table 2, we aimed to have all reclassification movements detailed, including cases, non-cases, and all UKB individuals. We think Table 2 is transparent in detailing this, as we show the number and proportions of individuals reclassified with the associated categorical NRI. The continuous NRI and IDI are detailed in Supplementary Table 19.

We have now edited the Discussion on pages 13–14 to read as follows to emphasize the difference between our preliminary, PRS-based approach and standard clinical risk scores, one main difference being the increased probability of false positives with the GRIT scores:

“The GRIT scores upclassified many non-cases to a higher risk at the thresholds aligned with the established clinical risk scores, but considering that they were derived to serve as preliminary risk indicators, the harms caused by these false positive classifications are minimal as the individuals would still require more detailed clinical risk assessment before possible preventative interventions.”

6: I think the message of the paper would be much clearer with a statement not just of NRI but of the number reclassified to achieve this net gain – something analogous to the number needed to treat.

Author response: We agree on the need for further discussion of the clinical utility of the GRIT scores, which is also detailed in answers to comments 1–5. Our approach to use categorical and continuous (category-free) net reclassification improvement (NRI) alongside integrated discrimination improvement (IDI) to measure predictive accuracy and clinical benefit is in line with previously published genetic risk prediction studies (e.g.,

PMIDs 33651632, 32273609, 32068818). As the goal was to develop a simple tool for preselecting individuals to further risk assessment, number needed to screen or number needed to treat -types of tools may perhaps not reflect the purpose of the tool. We have now added additional remarks about the classification of individuals in addition to the modifications made in response to comments 1–5 and the comment below. We have now modified the Discussion on page 14 to read as follows:

“As both the GRIT scores and the clinical risk scores failed to classify as high risk many of the individuals who had first disease events during the follow-up, continuous preventative efforts are crucial to combat the development of both lifestyle and clinical risk factors early on in younger individuals, particularly in those at high polygenic risk, to reduce the socioeconomic impact of cardiometabolic diseases on society.”

7: A focus on the performance of PRS solely in the high risk (young, before onset of the clinical factors; female, normoweight) should provide a stronger statement of utility. The data presented seem to reaffirm that PRS have little added utility in moderate or high clinical risk individuals.

Author response: We agree and this comment ties also into the answers to comments 1–6 about the clinical utility of the GRIT scores. We have updated the main Figure 2 (reviewer #2 comment 1 for reference), in which we detail the breakdown of the predictors (including PRS) included in the GRIT scores.

Based on this comment, we have now added a new Supplementary Figure 1, in which we detail the performance of the predictors across sex and age subgroups (below for reference). In CHD across these subgroups, the PRS_{CHD} combined with age and sex has a higher AUC compared to the conventional risk factors. In T2D across these subgroups, the PRS_{T2D} has a similar or higher AUC compared to the questionnaire-based risk factors (excluding BMI), but lower AUC compared to TG and HDL combined with age and sex.

Supplementary Figure 1. Area under the receiver operating characteristic curve (AUC) with 95% confidence intervals (CI) for individual risk factors and risk models in the validation cohort, UK Biobank for sex and age subgroups. Panel (a) shows results for CHD (N = 242,687) and panel (b) for T2D (N = 121,113). Points indicate AUC estimates and error bars represent the 95% CIs for each factor with incident disease as endpoint BMI = body mass index; LDL = low-density lipoprotein; HDL = high-density lipoprotein; TG = triglycerides; CVD = cardiovascular disease.

We have now modified the Discussion on page 14 to read as follows:

“While we observed improved performance in all included subgroups when adding supplemental risk factors to PRS-based risk models, the PRSs and the GRIT scores had the best discrimination in women and younger individuals, highlighting the performance of PRS-based risk tools in these groups of individuals in which clinical risk scores often have limited utility⁷.”

Minor Concerns:

8: Table 1: Numbers for Derivation CHD do not add up: 30,836+28,104 ≠61,878

Author response: Thank you for noticing this. The issue is now corrected in the revised manuscript in Table 1.

9: Does age refer to age at enrollment or incidence or endpoint?

Author response: Age refers to age at enrollment, and we have now clarified this in the revised manuscript in the legend of Table 1.

10: Page 9, Line 12: In the Statistical Analysis section, maximum follow-up time is stated as 10 years from baseline (enrollment?), but on the next page in the Results the median follow-up time was 15.3 years. Please explain.

Author response: In the Results the follow-up time was shown as the follow-up time before limiting the follow-up time of the study setting to 10 years from enrollment. This was because we assessed the performance of PRSs (Supplementary Table 10) using this follow-up. We have now clarified this in the manuscript and reported the overall median follow-up times in the legend of Supplementary Table 10.

We have now edited the Results on page 10 to show the median follow-up time and interquartile range after limiting the follow-up time to a maximum of 10 years after enrollment:

“The median follow-up time was 10.0 years (interquartile range [IQR], 7.8–10.0) for CHD and 10.0 years (IQR 7.5–10.0) for T2D.”

“Median follow-up was 10.0 years (IQR, 8.6–10.0) for CHD and 10.0 years (IQR, 8.3–10.0) for T2D.”

11: I understand the rationale for removing prevalent cases at Baseline, but can the authors estimate the impact on under-estimation of prevalence and PRS performance in a naïve (ideally, younger) population. The genetics should be stronger in the removed prevalent cases, but disease is also less prevalent.

Author response: In this study, we predicted incident, future disease cases and did not use prevalent cases which tend to have higher effect sizes and are biased by secondary

prevention. This approach is in line with previous risk prediction studies that similarly focus on predicting new disease cases. Genetic risk indeed has a stronger influence in the removed prevalent (and hence often younger) cases of CHD and T2D, despite the impact on prevalence in the study sample. This is detailed in Supplementary Table 16 (below for reference), the ORs per SD for the PRSs are higher for prevalent cases of CHD and T2D. As the goal of our risk tool is predict future cases (in line with established clinical risk calculators), we feel that further extending our analyses to prevalent disease cases would be outside the scope of the study.

PRS	Prevalent cases only		Incident cases only		All cases	
	AUC (95% CI)	OR per SD (95% CI)	AUC (95% CI)	OR per SD (95% CI)	AUC (95% CI)	OR per SD (95% CI)
PRS_{CHD}						
FinnGen	0.869 (0.867–0.871)	1.59 (1.57–1.62)	0.913 (0.911–0.916)	1.44 (1.41–1.47)	0.871 (0.869–0.873)	1.56 (1.53–1.58)
UK Biobank	0.811 (0.808–0.815)	1.77 (1.73–1.80)	0.756 (0.751–0.761)	1.61 (1.57–1.65)	0.792 (0.789–0.795)	1.72 (1.70–1.75)
PRS_{T2D}						
FinnGen	0.810 (0.808–0.813)	1.59 (1.57–1.61)	0.852 (0.849–0.855)	1.52 (1.49–1.55)	0.758 (0.756–0.761)	1.59 (1.57–1.61)
UK Biobank	0.725 (0.721–0.729)	1.75 (1.72–1.78)	0.669 (0.664–0.675)	1.51 (1.48–1.54)	0.708 (0.705–0.711)	1.68 (1.65–1.70)

Supplementary Table 16. AUCs and ORs per SD (with 95% CI) for our PRSs in FinnGen and the UK Biobank separately for prevalent, incident and all (prevalent and incident) disease cases in the full FinnGen dataset (N = 309,154) with 33,628 cases of CHD and 44,266 cases of T2D and the UK Biobank British ancestry subset (N = 343,672) with 18,698 cases of CHD and 24,192 cases of T2D. The estimates are from logistic regression models adjusted for year of birth, sex, and additionally ten first principal components of ancestry, batch, and genotyping array in FinnGen. The median follow-up time after enrollment in FinnGen was 15.3 years (interquartile range [IQR], 7.8–22.6) for CHD and 13.0 years (IQR 7.5–19.7) for T2D. The median follow-up after enrollment in UK Biobank was 10.7 years (IQR, 8.6–11.6) for CHD and 10.4 years (IQR, 8.3–11.3) for T2D.

12: Please provide more details on the calibration: do not assume that readers understand the concepts of baseline hazard and mean component, or how they were used for recalibration. This is really important in my opinion since calibration is critical but poorly discussed or understood in the literature.

Author response: We have now modified the Methods on page 9 to read as follows:

“We recalibrated all models except FINDRISC by using Cox proportional hazards models to estimate the baseline survival function and calculated the mean component (the sum of the predictor variable means weighted by respective coefficients) in the final UK Biobank validation datasets and combined these with the models’ published coefficients in the risk equations to obtain recalibrated 10-year risk estimates (recalibration parameters in Supplementary Table 8).”

13: Accordingly, I am concerned that the prevalence in FinnGen (Table 1) is ~3X greater for both traits in Finland than the UK. Quite apart from this being surprising to me, it must impact PRS assessment, but intuitively (correct me if I am wrong) in the opposite direction to that shown in Fig 2. Higher prevalence generally implies lower OR per Falconer, so applying the Finn OR to UKBB ought to underestimate prevalence, but apparently predicted risk is greatly elevated in the UKBB. Also, there does not appear to have been an adjustment for Finnish ancestry. These issues should be discussed for clarity.

Author response: The disease prevalence and incidence are higher in FinnGen as it is a collection of population-based and clinical biobank samples, broadly representative of the Finnish population. The UK Biobank is not representative of the UK population with known oversampling of healthy individuals, which has contributed to lower disease prevalence and incidence of CHD and T2D, though with somewhat less impact on the effect sizes of risk factors (PMID 32051121). We have now modified the Discussion on page 15 to read as follows:

“Despite a very low response rate resulting in oversampling of healthy individuals in UK Biobank, the risk factor associations have been considered to be generalizable³⁹”

We agree with the reviewer that the mentioned differences between the derivation and validation cohorts may impact the PRS-based assessment (see Supplementary Table 16 in the response to comment 11, the ORs per SD for the PRSs are higher in the UK Biobank compared to FinnGen). The differences in incidence/prevalence generally has an effect particularly on absolute risks and model calibration, and to a lesser extent on relative risks. The higher AUC of the PRS models in FinnGen is also partly explained by the adjustment for technical covariates (batch and genotyping array), and birth year (which may also capture period effects).

We have now updated the derivation analyses to be adjusted for the first 10 principal components of Finnish ancestry. All our results remained highly consistent after this update. We have now modified the Methods on page 7 to read as follows:

“To derive the risk tools, we used a Cox proportional hazards model to estimate beta coefficients, baseline hazard and mean component, adjusting for the first 10 principal components of Finnish ancestry and stratified the analyses by sex”

14: Reproducibility of methods is fine, but as I understand it few researchers have open access to Finnish databanks - though it can be arranged as a collaboration.

Author response: We share the reviewer’s concern about access to Finnish databanks, however, the Finnish biobanks offer global access. The authors who performed the data analyses have been granted access to the FinnGen study data which contains a collection of Finnish biobank data. The FinnGen data may be accessed through Finnish Biobanks’ FinBB portal (web link: www.finbb.fi, email: info.fingenious@finbb.fi). The Finnish biobank data can also be accessed through the Fingenious® services (<https://site.fingenious.fi/en/>) managed by FINBB.

15: I do not feel that all of the Supplementary Data is necessary, so in that sense they have gone overboard in promoting repeatability.

Author response: We agree and are concerned about the amount additional information in the Supplementary Information. However, we decided to catalog in detail the technical details of the study for transparency, to aid repeatability for readers with a specific interest in how the models were derived, and as a reference for future studies, also in line with the requirements for *Communications Biology*.

16: Depositing the PRS-CS scores in the public database on publication is really good.

Author response: Thank you, we agree and feel that this is important for the field. Therefore all polygenic scores will be deposited to the PGS Catalog prior to acceptance.

Reviewer comments, second version:

Reviewer #1 (Remarks to the Author: Overall significance):

The authors have adequately addressed my concerns. I think this is a very interesting manuscript targeted to a particular and important goal in population health management - can we use genetics plus a simple survey to identify people for further clinical work-up and intervention? The manuscript clearly demonstrates this is possible, does a good job providing re-classification metrics that drive this point home. Whether the approach is marginally better or worse than QRISK etc is an important comparison to present but ultimately the relative performance (slightly better or worse) is somewhat unimportant - it is approximately equivalent, simple to generate, and provides useful re-classification of risk for further follow-up. Novelty is somewhat low, but practicality and utility is high.

- Ali Torkamani PhD

Reviewer #1 (Remarks to the Author: Impact):

I am sure that this paper will be of high impact in Finland and drive some of their future directions. Whether or not there is an impact outside of the country would remain to be seen - but I think it lays out an interesting concept that would likely need to be modified for application elsewhere. In the USA, where healthcare is more individualistic, I'm not sure the more simple approach would take off as population health management is a bit of a foreign concept overall, though you could imagine for particular systems like a Kaiser system, this approach could be useful.

Reviewer #1 (Remarks to the Author: Strength of the claims):

As described above, the primary finding is equivalent performance generated in a simple manner with interesting risk re-classification. As a screening tool, the claims are appropriate.

Reviewer #1 (Remarks to the Author: Reproducibility):

I think that a comment on the eurocentric nature of the dataset should be made. It is there, but not very strongly stated. The PRS-CS approach, while powerful, in my opinion increases the risk of lack of generalizability outside of European populations. For me, a stronger caveat than "External validation of the GRIT-CHD and GRIT-T2D 22 scores is needed for diverse ancestries, including integration of ancestry-specific PRSs" is appropriate.

Reviewer #2 (Remarks to the Author: Overall significance):

The authors have addressed all my concerns and made the necessary changes to the manuscript.

Reviewer #3 (Remarks to the Author: Overall significance):

Reviewer 1 and I seemed to share concerns over novelty, but agreed that this study represents the state of the art and an important statement regarding the state of PRS for the two most prevalent common diseases. I think the confusion around "survey results" could perhaps be further clarified in the Intro or Discussion (as opposed to Methods) with a more direct statement of what factors go into GRIT and the comparable non-PRS models. Thus, for CHD, GRIT includes age, sex, smoking, diabetes status, BP meds, BMI and family history, all of which are surveyable and all but the last are in QRISK3 (BMI needs to be added to the list in the Methods). So, GRIT performs as well as QRISK3, but when you add non-survey measures that are part of QRISK3 (BP, cholesterol) it does even better. Since these gains, at least for CHD, are greater in younger individuals, I would strongly advocate for inclusion of the new Supplementary Figure 1 as a Figure in the main paper.

Author rebuttal, second version:

Please address the following concerns from Reviewers #1 and #3, and provide a brief cover letter + tracked-change version of the manuscript outlining where these edits have been made:

Referee #1:

I think that a comment on the eurocentric nature of the dataset should be made. It is there, but not very strongly stated. The PRS-CS approach, while powerful, in my opinion increases the risk of lack of generalizability outside of European populations. For me, a stronger caveat than "External validation of the GRIT-CHD and GRIT-T2D 22 scores is needed for diverse ancestries, including integration of ancestry-specific PRSs" is appropriate.

Referee #1: We agree with the need to further state the risk of lack of generalizability outside of European populations better. We have now more strongly stated this limitation in line with the requested change. Discussion on page 10 lines 5-9 now reads: "Additionally, our data was limited to middle-aged individuals of European ancestry, which increases the risk of lack of generalizability outside of European populations. Expanding validation of the GRIT-CHD and GRIT-T2D scores to more diverse populations is needed before possible clinical implementation, including integration

Referee #3:

I think the confusion around "survey results" could perhaps be further clarified in the Intro or Discussion (as opposed to Methods) with a more direct statement of what factors go into GRIT and the comparable non-PRS models. Thus, for CHD, GRIT includes age, sex, smoking, diabetes status, BP meds, BMI and family history, all of which are surveyable and all but the last are in QRISK3 (BMI needs to be added to the list in the Methods).

of ancestry-specific PRSs to improve applicability across ethnic groups”.

Referee #3: We have now further clarified the Discussion make a direct statement of all the risk factors that are included in the GRIT scores and that overlap with the clinical risk scores, in addition to the statements in the Methods and the presentation in Figures 2 & 3. Most of the risk factors included in GRIT are also included in the clinical risk scores, with some exceptions. We further state that the GRIT scores did not contain all risk factors that the clinical risk scores included. This should impact some aspects of the risk performance, as we have previously noted in the discussion. Please see the edits in Discussion page 8 lines 7-19:

“In addition to PRS_{CHD}, the GRIT scores for CHD include age, sex, smoking status, BMI, blood-pressure-lowering medication use, history of diabetes, and family history of CHD and additionally SBP, high-density lipoprotein [HDL], and low-density lipoprotein [LDL] in GRIT-CHD+, all of which are also included in QRISK3 and mostly in PCE (both algorithms use total cholesterol [TC] instead of LDL). Similarly, in addition to PRS_{T2D}, the GRIT

scores for T2D include age, sex, BMI, smoking status, current blood-pressure-lowering medication use, current statin use, history of CVD, history of gestational diabetes, and family history of diabetes and additionally SBP, triglycerides [TG], and HDL in GRIT-T2D+, all of which (except SBP and lipid measurements) are also included in QDiabetes and most of which also in FINDRISC. The clinical risk scores also include components we could not include in the GRIT scores due to data limitations, such as measures for diet, physical activity, socioeconomic factors, and waist circumference, but their effects may be mediated to an extent by risk factors that were included, such as smoking and BMI.”, and Methods page 15 lines 6-7: “QRISK3 also includes BMI, additional comorbidities, as well as socioeconomic risk factors.”